# Alpha-synuclein is strategically positioned for afferent modulation of midbrain dopamine neurons and is essential for cocaine preference

Olga Trubetckaia[1], Ariana E. Lane[1], Liping Qian[1], Ping Zhou[1] & Diane A. Lane[1]*

Alpha-synuclein (α-syn) is an abundant neuroprotein elevated in cocaine addicts, linked to drug craving, and recruited to axon terminals undergoing glutamatergic plasticity - a proposed mechanism for substance abuse. However, little is known about normal α-syn function or how it contributes to substance abuse. We show that α-syn is critical for preference of hedonic stimuli and the cognitive flexibility needed to change behavioral strategies, functions that are altered with substance abuse. Electron microscopic analysis reveals changes in α-syn targeting of ventral tegmental area axon terminals that is dependent upon the duration of cocaine exposure. The dynamic changes in presynaptic α-syn position it to control neurotransmission and fine-tune the complex afferent inputs to dopamine neurons, potentially altering functional dopamine output. Cocaine also increases postsynaptic α-syn where it is needed for normal ALIX function, multivesicular body formation, and cocaine-induced exosome release indicating potentially similar α-syn actions for vesicle release pre- and post-synaptically.

[1] Feil Family Brain and Mind Research Institute, Weill Cornell Medical College, New York, NY 10065, USA. *email: dal2021@med.cornell.edu

Compulsive drug use and unremitting craving even in the absence of drugs are some of the hallmarks of substance abuse disorders, making this disorder extremely difficult to overcome[1,2]. These relatively permanent changes in behavior have been attributed to drug-induced maladaptive glutamatergic plasticity of dopamine neurons in the ventral tegmental area (VTA), neurons responsible for reward and motivated behaviors[3,4]. The circuitry within the VTA is complex and dopamine activity is a balance of excitatory and inhibitory inputs coming from multiple brain regions[5–9]. Drugs of abuse influence this afferent circuitry by changing VTA dopamine activity from a steady tonic rate of firing, which is under GABAergic (γ-aminobutyric acid) control, to high-frequency bursting, which is mediated by increased glutamate AMPA (α-amino-3-hydroxy-5-methyl-4-isoxazolepropionic acid) receptor activation and is necessary for long-term potentiation of dopamine neurons[10–14]. However, our understanding of the mechanisms that mediate the shift in glutamate activation of dopamine neurons and how those changes lead to decreasing reward and increased compulsion is incomplete.

In humans, repeated cocaine use elevates α-synuclein (α-syn) levels in the blood and brain tissue[15,16], which is correlated with increased cocaine craving[16]. Moreover, α-syn is a ubiquitous neuroprotein that is upregulated in brain regions undergoing developmental neuroplasticity[17]. There is growing evidence that α-syn acts presynaptically to regulate neurotransmission; however, the findings, especially in regards to dopamine activity, are mixed with support for both facilitatory[18–21] and inhibitory effects[22–24]. Nevertheless, these studies highlight the importance of dynamic pre-synaptic α-syn expression for changes in synaptic neurotransmission, especially pertaining to dopamine activity.

The reported upregulation of α-syn in regions undergoing neuronal plasticity and cocaine-addicted individuals prompted our investigation of cocaine-mediated behavioral and subcellular α-syn changes in normal C57BL/6J and α-syn knockout (KO) (C57BL/6N-Snca^{tm1Mjff}/J) mice. We show that α-syn is critical for the cognitive flexibility needed to produce changes in motivated behaviors, specifically towards rewarding stimuli. Also, α-syn targets specific afferent fibers making synaptic contacts onto VTA dopamine neurons, potentially influencing regional inputs and activity of dopamine neurons. Finally, we demonstrate previously undescribed post-synaptic actions of α-syn where it is important for normal multivesicular body (MVB) formation and cocaine-mediated exosome release.

## Results

α-Syn KO and wild-type (WT) mice showed similar increases in locomotor activity and sensitization to cocaine across trials. No significant differences in locomotor activity were evident at baseline ($F(1,32) = 0.98$, $p > 0.05$), with initial cocaine injection ($F(1,19) = 2.24$, $p > 0.05$), or in the development of locomotor sensitization to repeated cocaine administration ($F(6,168) = 13.25$, $p = 0.001$; Fig. 1a). In contrast, α-syn KO mice did not demonstrate preference or aversion to cocaine-paired environments, unlike WT controls who spent significantly more time in the cocaine-paired environments ($F(1,32) = 17.22$, $p = 0.001$; Fig. 1b).

The lack of cocaine-conditioned place preference (CPP) in α-syn KO mice was surprising given that these mice showed locomotor sensitization demonstrating a progressive activation of dopamine neurons across cocaine trials. This discrepancy could reflect deficits in associative memory or overall changes in motivation and/or preference for cocaine. To evaluate potential contributing factors, spatial memory and the consumption of highly palatable sweetened condensed milk (SCM) were assessed.

There were no significant differences in spatial memory acquisition (days 1–4) between α-syn KO and WT mice (Fig. 2a).

All mice demonstrated similar acquisition rates with a progressive decrease in the latency to find the escape hole ($F(7,98) = 11.34$, $p = 0.001$) and entry errors ($F(7,98) = 6.91$, $p = 0.001$) across acquisition trials. No significant difference in genotypes was seen in the latency to find the escape hole ($F(1,14) = 2.40$, $p > 0.05$) or number of errors ($F(1,14) = 0.39$, $p > 0.05$; Fig. 2a).

On D5, the position of the escape hole was moved to assess the ability of the mice to change strategies and timing to learn a new escape position (Fig. 2b). The prior learning trials (D1–4) interfered with the novel learning task (D5) in α-syn KO mice who took longer to find the new escape hole position ($F(1,14) = 4.97$, $p = 0.04$) and made significantly more entry errors than WT mice ($F(1,14) = 5.52$, $p = 0.03$). Moreover, α-syn KO mice demonstrated greater perseverative errors than WT mice. They had increased attempts to enter the original escape hole position ($F(1,14) = 6.63$, $p = 0.02$) and spent a greater amount of time in the quadrant containing the original escape hole position ($F(1, 14) = 18.94$, $p = 0.001$; Fig. 2b).

Both α-syn KO and WT mice ingested more SCM than water indicating a preference for SCM ($F(6,168) = 19.69$, $p = 0.001$; Fig. 2c). However, the predilection for SCM was not as strong in α-syn KO mice, which demonstrated diminished SCM intake as

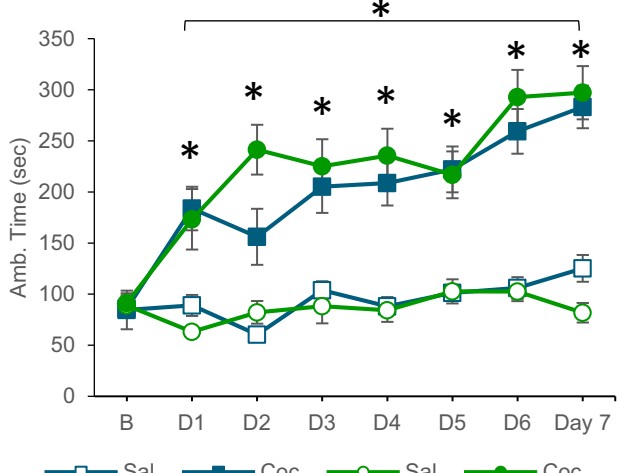

**A** Locomotor Activity

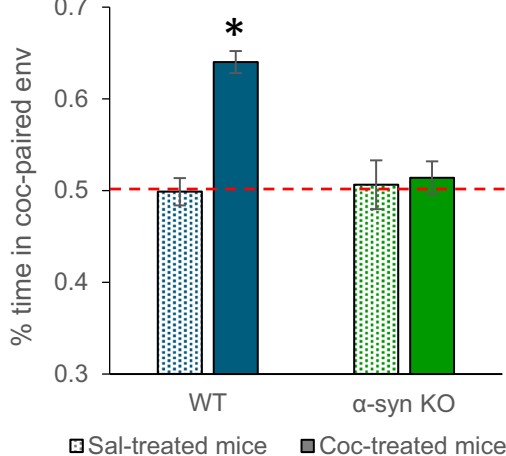

**B** Cocaine Conditioned Place Preference

**Fig. 1 a** Both α-syn KO (green) and WT (blue) mice show progressive increases in ambulatory time (s) across cocaine inj. trials. **b** Lack of conditioned preference to cocaine in α-syn KO mice (green bars). *$P < 0.05$

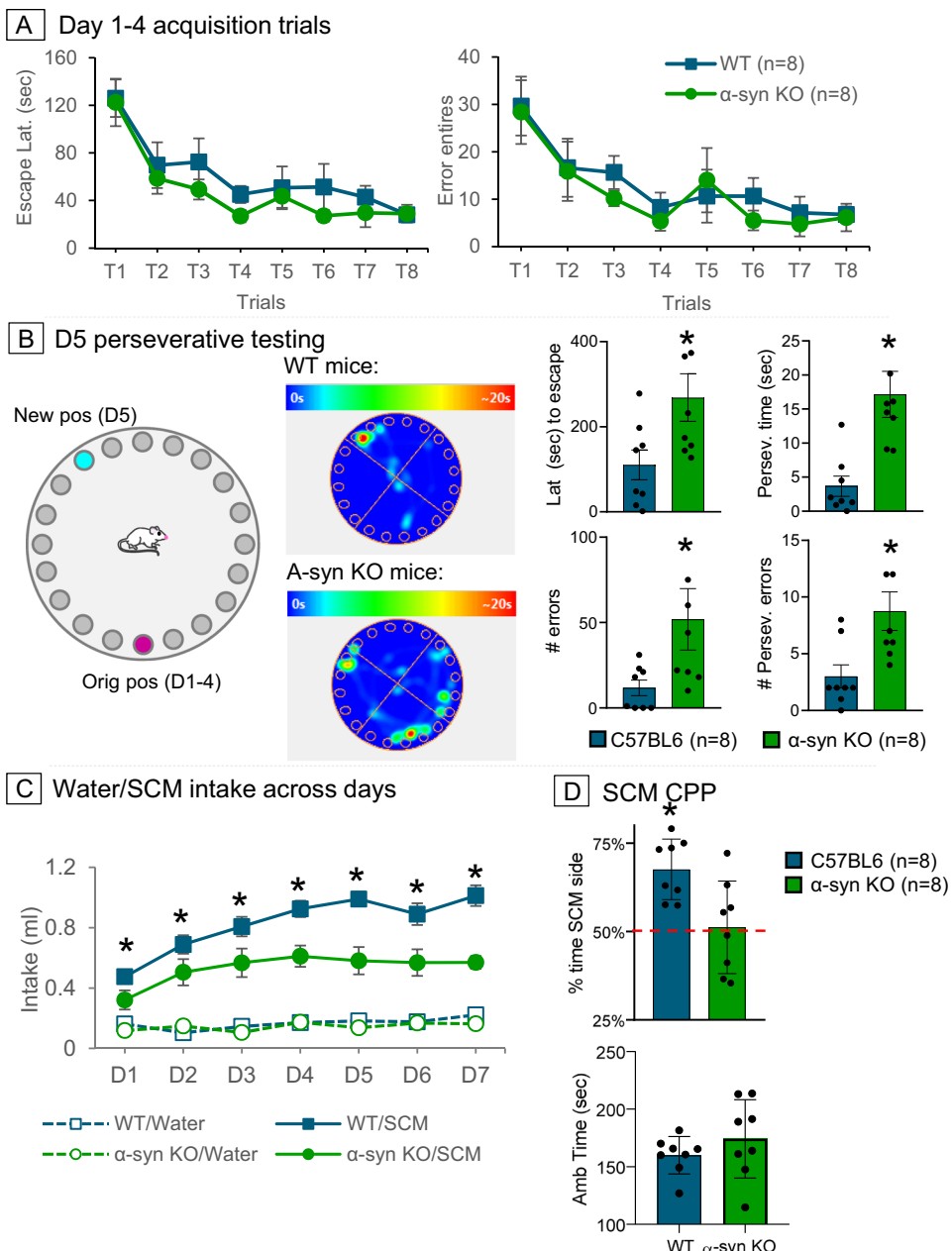

**Fig. 2 a** α-Syn KO (green) and WT (blue) mice show similar latencies to find the escape hole and number of errors made during acquisition of the Barnes Maze spatial memory task. **b** Schematic drawing of the Barnes Maze with the original escape hole position at 6 o'clock (magenta) and changed position (teal) at ~11 o'clock for perseverative testing. Heat map images of mouse location during D5 perseverative testing showing that α-syn KO mice spent more time near the original escape hole position than WT mice. Bar graphs showing that α-syn KO mice took longer to find the newly positioned escape hole, had more general and perseverative errors, and spent more time in the quadrant with the original escape hole position than WT mice. **c** During the conditioning trials (D1–7), α-syn KO and WT mice ingested more SCM (solid lines) than water (dashed lines), but to a lesser extent by the α-syn KO mice. In addition, α-syn KO mice did not show conditioned preference to SCM; *$p < 0.05$, SCM = sweetened condensed milk

compared to WT mice ($F(6,84) = 16.95$, $p = 0.001$). In addition, α-syn KO mice had a different pattern of SCM ingestion than WT mice. Their SCM intake plateaued at D4, whereas WT mice continued to increase their intake across trials ($F(6,84) = 3.25$, $p = 0.01$; Fig. 2c). Consistent with the reduced preference for SCM, α-syn KO mice did not show a CPP or aversion to a SCM-paired chamber ($t(14) = 2.96$, $p = 0.01$; Fig. 2d). To determine whether the decreased SCM intake in the α-syn KO mice influenced the expression of CPP, a correlation between SCM intake and the amount of time spent in the SCM-paired chamber was conducted. WT mice show a positive correlation between SCM

intake and side preference ($r = 0.71$, $p = 0.001$). However, no relationship exists in α-syn KO mice ($r = 0.30$, $p > 0.05$), indicating that the amount of SCM intake is not a critical factor for the lack of CPP in α-syn KO mice. The SCM findings parallel the cocaine place-conditioning experiments (where all animals received the same amount of cocaine), revealing a diminished preference for rewarding stimuli in α-syn KO mice.

Immunoblot analyses of midbrain lysates from WT mice showed increased α-syn protein levels after cocaine injections (Fig. 3), showing a two-fold increase in α-syn as compared to saline controls following repeated cocaine ($F(2,10) = 7.29$; $p =$

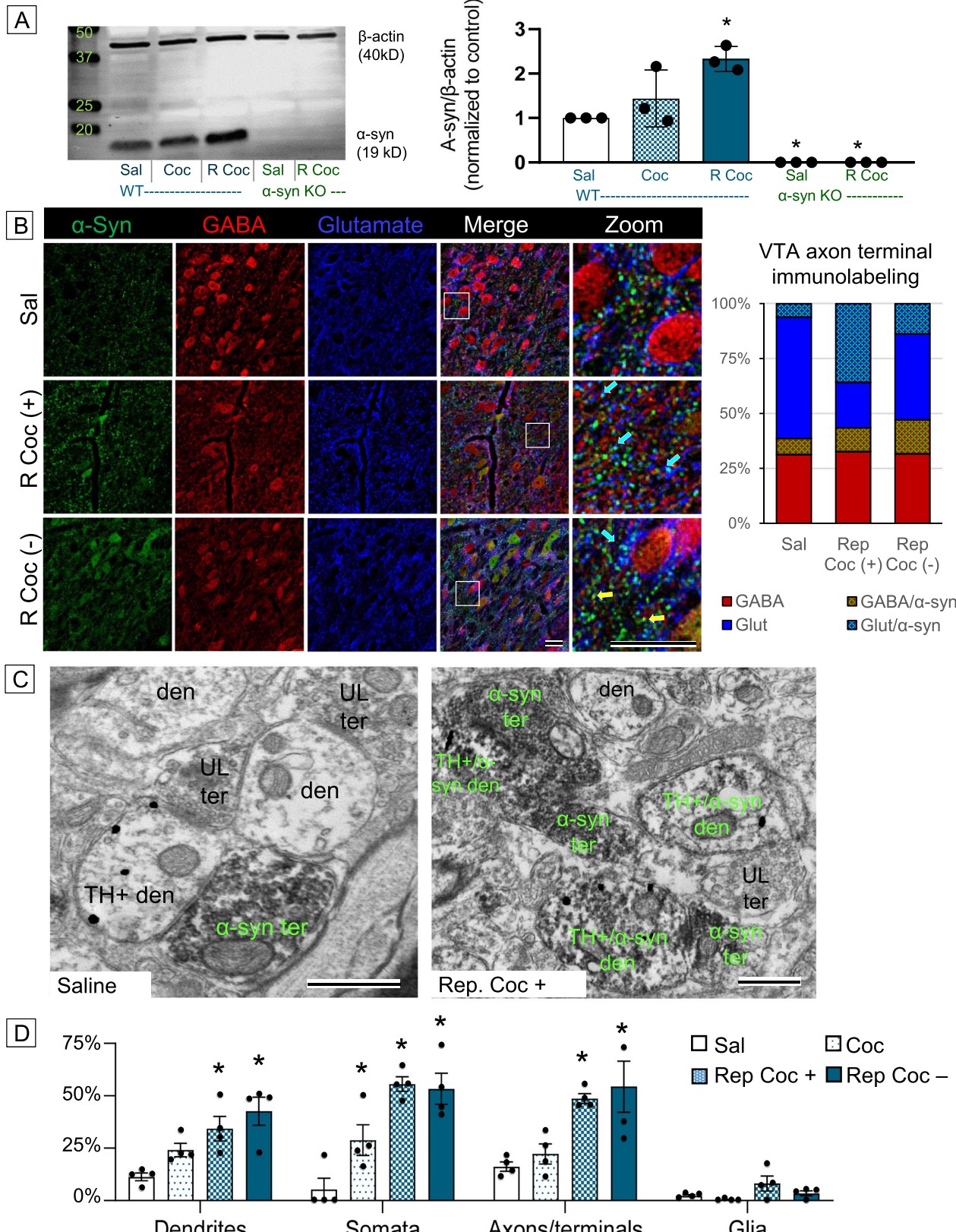

**Fig. 3 a** Western blot of α-syn and quantification showing increased α-syn protein levels in the midbrain after cocaine administration.) Confocal images of VTA tissue and quantification of labeled axon terminal puncta (bar graph) showing increased co-labeling for α-syn and glutamate (teal puncta/bar) when cocaine is systemically present, but increased co-labeling for α-syn and GABA (yellow puncta/bar) when cocaine is systemically absent after repeated administration. Scale bar = 25 μm. **c** Electron micrographs of VTA tissue from saline- and repeated cocaine-treated mice showing that cocaine increases both pre- and postsynaptic α-syn immunolabeling (green labels); scale bar = 500 nm. **d** Proportion of neuronal and glial profiles showing increasing α-syn immunolabeling after cocaine; *$p < 0.05$

0.01). No discernable α-syn immunolabeling was evident from α-syn KO mice demonstrating specificity of the α-syn antibody (lanes 4, 5; Fig. 3a).

Confocal microscopy shows that cocaine-naive mice (sal) had relatively sparse and almost exclusively pre-synaptic α-syn immunolabeling (Fig. 3b). Triple labeling for α-syn (green), GABA (red), and glutamate (blue) in the VTA revealed that the proportion of α-syn immunolabeling was roughly equal between GABA and glutamate co-labeled axon terminal puncta in saline-treated mice (7% and 6%, respectively). Repeated cocaine, when systemically present (rep coc+), increased in α-syn labeling of axon terminal puncta, especially in those co-labeled for gluta-mate. Tissue process 72 h after the last cocaine injection (rep coc−) also showed increased α-syn immunolabeling, but had proportionately more α-syn labeling in axon terminal puncta co-labeled with GABA, demonstrating a shift in α-syn localization that is dependent on ongoing systemic cocaine activity ($\chi^2(9) = 4.29$, $p = 0.03$).

α-Syn immunolabeling was also evident post-synaptically after cocaine. Cell bodies containing α-syn immunolabeling alone or together with GABA were evident in both the rep coc+ and rep coc− groups (Fig. 3b, middle and bottom panels). Because dopaminergic and GABAergic are the two main cell types found within the VTA[25] and no somata with glutamate immunolabeling were seen in the micrographs, the findings support the idea that both cell types have an upregulation of α-syn after repeated cocaine administration. This was confirmed with subsequent electron microscopic (EM) analysis.

For a more precise and detailed characterization of subcellular changes of α-syn in identified neuronal and glial profiles, we utilized EM analysis of α-syn and tyrosine hydroxylase (TH) immunolabeling within the VTA. The immunolabeling was combined with morphological identification of neuronal/glial profiles in the neuropil, allowing for in-depth analysis of cocaine-mediated changes in α-syn subcellular localization. Like confocal imaging, electron micrographs show that cocaine-naive mice had relatively low levels of α-syn immunolabeling, which was pre-dominantly pre-synaptic (Fig. 3c, left). After repeated cocaine administration, α-syn immunolabeling was much more prevalent and seen in both pre- and postsynaptic profiles within the neuropil (Fig. 3c, right). Quantification of labeled profiles shows that <10% of total neuropil profiles have α-syn labeling in cocaine-naive mice. Of those, nearly two-thirds are axon terminals (Fig. 3d). A single cocaine injection did not significantly change pre-synaptic α-syn amounts; however, it did increase postsynaptic α-syn immunolabeling. The proportion of both pre- and post-synaptic neuronal profiles with α-syn immunolabeling progres-sively increased with repeated cocaine administration, where roughly 50% of neuronal profiles contained α-syn immunola-beling regardless of whether cocaine was systemically present (rep coc+) or not (rep coc−; $F(3,44) = 26.06$, $p = 0.001$; Fig. 3d). α-Syn immunolabeling was rarely seen in glial profiles; however, repeated cocaine did increase glial immunolabeling, as compared to saline controls.

Synaptic contacts between axon terminals and dendrites/somata can be characterized by their morphological structure with the extreme magnification levels supported by electron microscopy. Asymmetric synapses (Fig. 4a; blue arrows) are known to make excitatory-type contacts because they are apposed to glutamatergic terminals[26] and associated with excitatory pro-teins[27]. Symmetric synapses (Fig. 4a; red arrows) make inhibitory-type contacts because they appose GABAergic terminals[28,29] and are associated with inhibitory proteins[27]. We exploited this morphological difference and combined our ana-lysis with immunolabeling to precisely characterize changes in α-syn following cocaine administration (Fig. 4). This technique

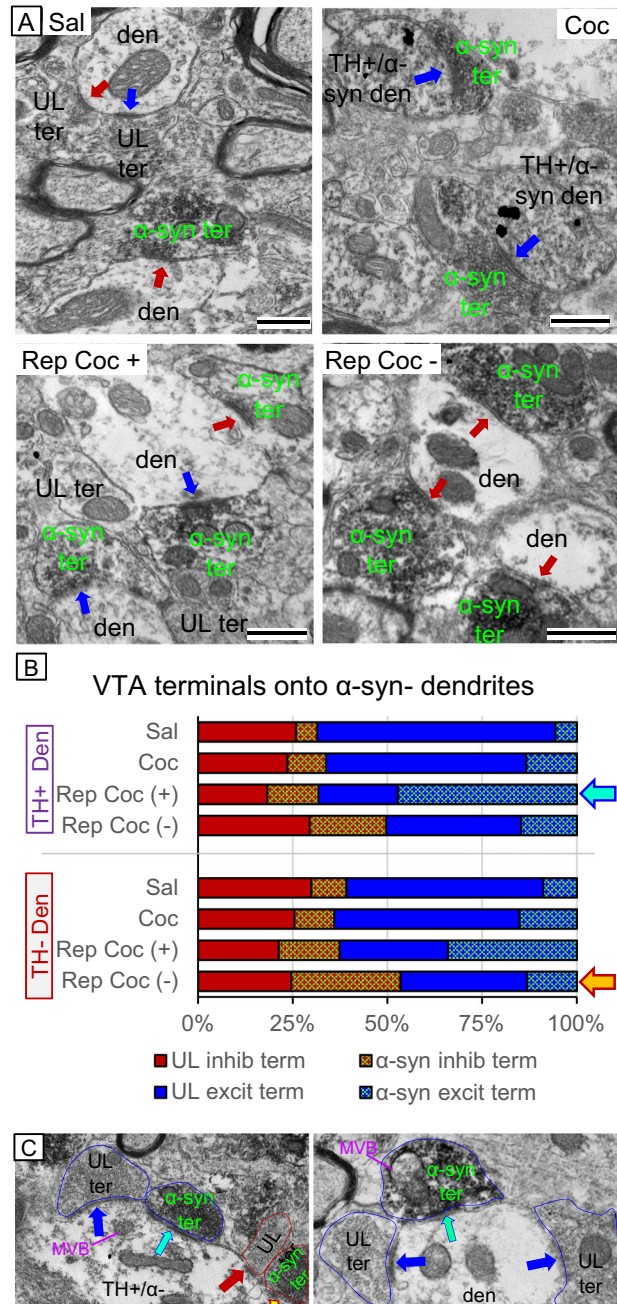

**Fig. 4 a** Electron micrographs showing axon terminals containing α-syn peroxidase immunolabeling (green labels) making excitatory (blue arrows) and inhibitory (red arrows) contacts onto VTA dendrites. **b** Quantification of characterized synaptic contacts onto TH+ and TH− dendrites in the VTA showing that when systemically present, repeated cocaine (rep coc+) increased α-syn immunolabeling in axon terminals making excitatory-type synaptic contacts onto dopamine dendrites (teal arrow). When cocaine or its metabolites were no longer systemically present (rep coc−), α-syn immunolabeling was increased in axon terminals making inhibitory-type synaptic contacts onto non-dopaminergic dendrites in the VTA (yellow arrow). **c** Electron micrographs showing the selectivity of α-syn immunolabeling. Axon terminals with (teal arrows) and without (blue arrows) α-syn immunolabeling can be seen making excitatory-type synaptic contacts onto the same dopamine (TH+) or non-dopamine dendrite within the VTA. Similarly, axon terminals with (yellow arrow) or without (red arrow) α-syn labeling can be seen making inhibitory contacts onto the same dopamine dendrite. Scale bar = 500 nm; den = dendrite; ter = axon terminal; UL = unlabeled

allowed for highly specific characterization of α-syn subcellular distributions, not only by the type of synaptic contact (inhibitory or excitatory) but also whether these axon terminals were contacting TH-positive (dopamine) or TH-negative (mostly GABA, but heterogeneous) dendrites within the VTA and whether the targeted dendrites contained α-syn or not.

**Axon terminals making contacts onto VTA dendrites without α-syn immunolabeling.** In saline controls, α-syn immunolabeling was relatively sparse and equally distributed within inhibitory and excitatory axon terminals. In axon terminals making contact onto TH+ dendrites, ~6% of the terminals (either inhibitory- or excitatory-type synaptic contacts) were immunopositive for α-syn (Fig. 4b). This amount was slightly higher in axon terminals making contacts onto non-TH dendrites, with 8% of inhibitory and 9% of excitatory axon terminals containing α-syn (Fig. 4b).

These ratios increased slightly with a single injection of cocaine; however, repeated cocaine (rep coc+) produced a three-fold increase in the number of excitatory axon terminals labeled with α-syn (teal arrow, Fig. 4b). This increase was most prevalent in axon terminals making contacts onto TH+ dendrites ($\chi^2(9) = 129.27$, $p = 0.001$), but also occurred with contacts onto TH− dendrites. When cocaine was no longer systemically present (rep coc−), the number of excitatory-type axon terminals with α-syn dramatically decreased. In contrast, the number of inhibitory-type axon terminals with α-syn immunolabeling had a two-fold increase. This was especially evident in inhibitory contacts onto TH− dendrites ($\chi^2(9) = 206.95$, $p = 0.001$; yellow arrow, Fig. 4b).

**Axon terminals making contacts onto VTA dendrites with α-syn immunolabeling.** Similar patterns in α-syn distributions were seen in axon terminals that made contacts with dendrites also containing α-syn immunolabeling, but to a lesser extent (TH+ dendrites: $\chi^2(9) = 24.82$, $p = 0.01$; TH− dendrites: $\chi^2(9) = 14.94$, $p = 0.09$; Supplementary Fig. 1). It is currently unclear why the post-synaptic presence of α-syn may affect the levels of α-syn in pre-synaptic contacts, but it points to possible α-syn-mediated signaling between cells that may be a function of increase release of extracellular vesicles (EVs) by cocaine (see below).

Cocaine-mediated α-syn upregulation was not global to all VTA axon terminals. Instead, the changes in α-syn immunolabeling were highly selective and targeted to specific axon terminals. For instance, when examining individual dendrites, axon terminals with α-syn immunolabeling making excitatory synaptic contacts (teal arrows) were adjacent to unlabeled axon terminals making excitatory synaptic contacts (blue arrows) onto the same dopamine dendrite (Fig. 4c). Similarly, inhibitory-type axon terminals with α-syn (orange arrows) are seen adjacent to inhibitory axon terminals without α-syn (red arrows) making synaptic contacts onto the same dopamine dendrite (Fig. 4c), indicating a functional targeting of α-syn to particular afferent VTA fibers.

Repeated cocaine also increased postsynaptic immunolabeling of α-syn in VTA dendrites and somata (Fig. 3d). The postsynaptic α-syn labeling was most evident on MVBs, both on their outer membrane and on intraluminary vesicles (magenta arrow), in intracellular vesicles (green arrows), and mitochondria (red arrows; Fig. 5a). The association of α-syn with MVBs and intracellular vesicles is of interest because cocaine administration significantly decreased the presence of MVBs in both TH+ and TH− dendrites in the VTA of WT mice (TH+: $F(3,14) = 9.12$, $p = 0.002$; TH−: $F(3,14) = 11.60$, $p = 0.001$), an effect that was attenuated in α-syn KO mice (TH+: $(F(3,15) = 0.62$, $p = 0.61$; TH−: $F(3,15) = 2.12$, $p = 0.14$; Fig. 5b).

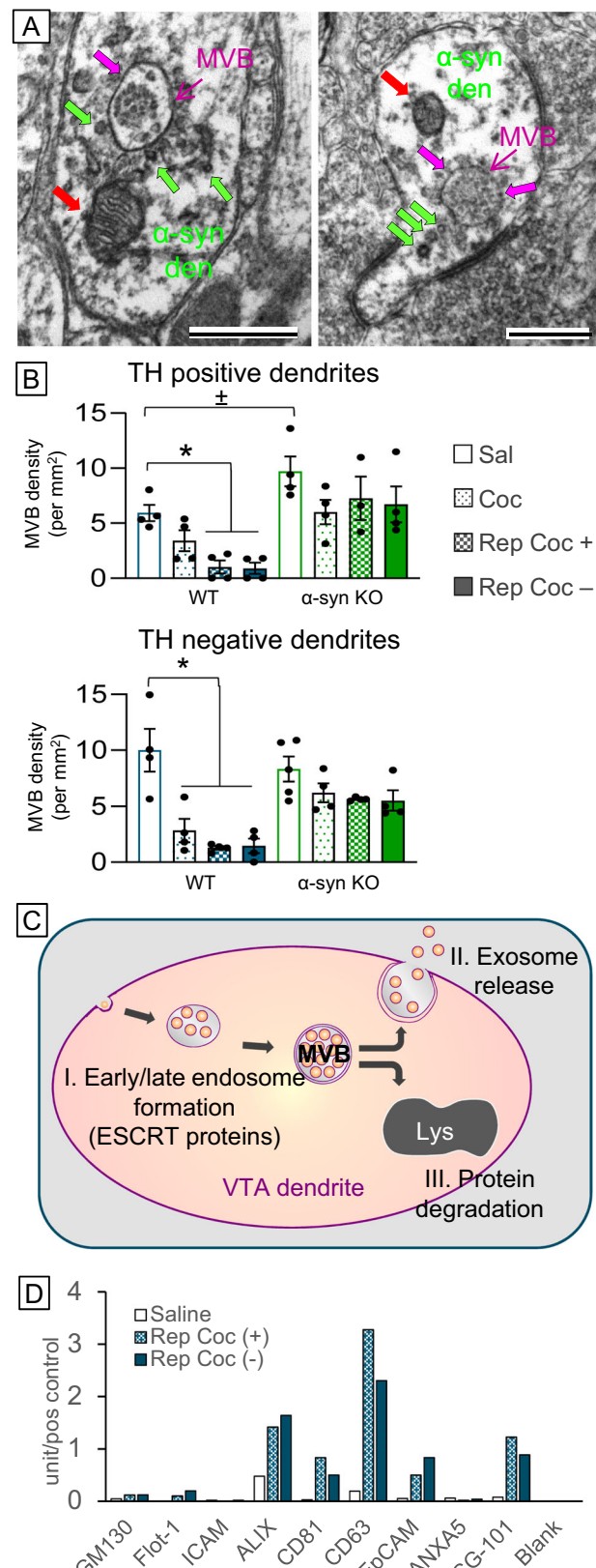

MVBs are organelles that can intake a large variety of biologically active protein, encapsulated in vesicles that can either be released as exosomes into the extracellular space for cellular communication or shuttled to lysosomes for degradation (Fig. 5c). Formation of MVBs is dependent on a family of endosomal sorting complexes required for transport proteins (ESCRT 0–III),

**Fig. 5 a** Electron micrographs showing postsynaptic α-syn immunoperoxidase labeling on intracellular vesicles (green arrows) and around the outer membrane of MVBs (magenta arrows) and mitochondria (red arrows) in VTA dendrites. Scale bar = 500 nm. **b** Bar graphs showing that repeated cocaine administration decreased the presence of multivesicular bodies in dopamine (TH+) and non-dopamine (TH−) dendrites within the VTA, an effect that was diminished in α-syn KO mice (green bars). **c** Schematic diagram showing that decreases in MVBs can be caused by cocaine-mediated (I) dysfunction of MVB formation by ESCRT protein machinery, (II) increased exosome release, or (III) increased protein degradation by lysosomes. **d** Bar graph of antibody array quantification of isolated serum exosome samples from saline- and cocaine-treated mice showing increased ALIX, CD63, and TSG-101 immunoreactivity with repeated cocaine administration

machinery necessary for vesicle invagination, formation, and membrane abscission. The decreased presence of MVBs witnessed in VTA dendrites after repeated cocaine administration may be a function of: decrease in MVB formation, increase in exosome release, or increase in lysosomal degradation of MVBs.

To narrow potential mechanistic targets for these functions, an exosome antibody array was performed on extracullular vesicles (EVs) isolated from blood samples from mice that received repeated cocaine. Cocaine elevated levels of apoptosis-linked gene 2-interacting protein X (ALIX), CD63, and tumor susceptibility gene 101 (TSG-101; $\chi^2(14) = 127.19$, $p = 0.001$; Fig. 5d), which were targeted in subsequent studies.

Western blot analyses of lysates obtained from midbrain tissue showed significant increases in ALIX and CD63 protein levels after repeated cocaine ($F(3,23) = 3.85$, $p = 0.02$ and $F(3,23) = 6.11$, $p = 0.01$, respectively; Fig. 6a). The cocaine-mediated increase in ALIX and CD63 was blocked in α-syn KO mice ($F(1,23) = 3.94$, $p = 0.05$ and $F(1,23) = 10.97$; $p = 0.001$), suggesting that the presence of α-syn is critical for the cocaine-induced changes in these proteins.

ALIX interacts with ESCRT-I and ESCRT-III complex subunits that are essential for the formation and sorting of intraluminal vesicles (ILVs) into MVBs[30–33]. Because the cocaine-mediated increase of ALIX was blocked in α-syn KO mice, we examined whether α-syn presence had an effect on MVB size and vesicle density within MVBs. Although cocaine decreased the number of MVBs (Fig. 5b) present in VTA dendrites, there were no significant differences in MVB size either with cocaine administration ($F(3,224) = 0.20$, $p = 0.90$) or between α-syn mouse genotypes ($F(3, 224) = .25$, $p = 0.86$). However, the absence of α-syn profoundly diminished the number of ILVs found within individual MVBs ($F(1,236) = 92.04$, $p = 0.0001$), irrespective of cocaine administration ($F(3, 236) = 1.06$, $p = 0.37$; Fig. 6b). The data indicate that α-syn is critical for ALIX-mediated formation of MVB ILVs.

ALIX also interacts with TSG-101, an ESCRT-I complex subunit[31,34]. Although serum-isolated EVs showed increased TSG-101 after cocaine (Fig. 5d), when examined specifically in the midbrain (Fig. 6a) there was no significant change in TSG-101 protein levels either with cocaine treatment ($F(1,15) = 0.99$, $p = 0.34$) or between α-syn mouse genotypes ($F(1,23) = 0.36$, $p = 0.55$). This indicates that TSG-101 does not play a role in the change in vesicle density seen in VTA neurons of α-syn KO mice, but may be affected in other systems.

CD63 is a tetraspanin located on the lipid membrane of EVs, as well as in late endosomes, MVBs, and lysosomes[35]. We found that repeated cocaine administration significantly decreased the presence of MVBs (Fig. 5b), but increased CD63 protein levels in the midbrain ($F(1,14) = 19.29$, $p = 0.001$; Fig. 6a), suggesting

that repeated cocaine administration may trigger exosome release in the VTA.

To investigate whether cocaine promotes exosome release, Western blot and EM analysis of EVs isolated from hemi-brain tissue were examined. CD63 immunoblots of isolated EVs showed significantly more CD63 immunoreactivity after repeated cocaine administration as compared to tissue obtained from cocaine-naive mice ($t(4) = 13.94$, $p = 0.001$; Fig. 6c), indicating increased EVs release with cocaine exposure. EVs isolated from the extracellular space from hemi-brain samples were also visualize and quantify using electron microscopy. The isolated vesicles were ~50–100 nm in diameter and showed the prototypic disc-like shape attributed to exosomes. Quantification of EVs in equal imaged fields shows a significant increase in the density of EVs (# of vesicles/square micron) after repeated cocaine administration ($t(25) = 5.00$, $p = 0.001$; Fig. 6d).

Subcellular CD63 immunolabeling of WT and α-syn KO mice was also characterized using electron microscopy (Fig. 6e–g). There was little CD63 immunolabeling in VTA neuronal and glial profiles of saline control mice (Fig. 6e left). When present, CD63 was evident around vesicle membranes and along the outer membrane of an MVB (magenta outline) within VTA dendrites (blue). In addition, although sparse, some CD63 immunolabeling was seen in the extracellular space (filled green arrow) between an axon terminal (red) and a glial process (Fig. 6e, left). However, most of the extracellular space was devoid of CD63 immunolabeling (open green arrows). After repeated cocaine administration, CD63 immunolabeling filled the extracellular space (filled green arrows; Fig. 6e right) between a dendrite (blue) and glial (yellow) cell in the VTA. Moreover, the glial cell (yellow) shows CD63 immunolabeling, suggesting that it may be taking up released EVs.

In both cocaine- and saline-treated mice, CD63 immunolabeling was evident around the outer membrane of MVBs (filled green arrow) and around vesicles (purple arrows) appearing to either be entering the dendrite/MVB or exiting the MVB into the extracellular space of a VTA dendrite (Fig. 6f). While cocaine decreased the number of MVBs within the VTA, it did not significantly impact CD63 immunolabeling of MVBs ($F(1,6) = 2.03$, $p = 0.20$). In contrast, MVBs in α-syn KO rarely showed CD63 immunoreactivity ($F(1,6) = 22.92$, $p = 0.003$). It is still unclear if this difference is a result of the lower vesicular density in these mice or if other factors influence CD63 levels.

Finally, CD63 immunolabeling was also highly expressed on the outer membrane (green arrows) of mitochondria (red) within the VTA (Fig. 6g). Repeated cocaine administration significantly increased the number of mitochondria positive for CD63 immunoreactivity ($F(1,6) = 40.85$, $p = 0.001$), an effect that was significantly attenuated in α-syn KO mice ($F(1,6) = 49.84$, $p = 0.001$; Fig. 6g).

To examine whether the cocaine-mediated decrease of MVBs is a function of increased lysosomal degradation, we examined immunoblot analysis of lysosomal associated membrane protein-1 (LAMP-1) protein levels from midbrain tissue lysates (Fig. 6a) and EM analysis of morphological characteristics of lysosomes within VTA somata (Fig. 6h) in WT and α-syn KO mice. Western blot analysis showed no significant changes in LAMP-1 protein levels either between cocaine treatment groups ($F(1,12) = 2.20$, $p = 0.16$) or by genotype ($F(1,12) = 2.61$, $p = 0.13$; Fig. 6a).

Because lysosomes (orange arrows, Fig. 6i) are conglomerates of protein degradation machinery, they generally do not have uniform size or shape within neurons. As such, EM analysis of individual lysosomal sizes and densities within a defined somatic profile (either TH+ or TH−) were measured with reference to a number of characteristics, including drug treatment, α-syn genotype, and the presence of α-syn immunolabeling.

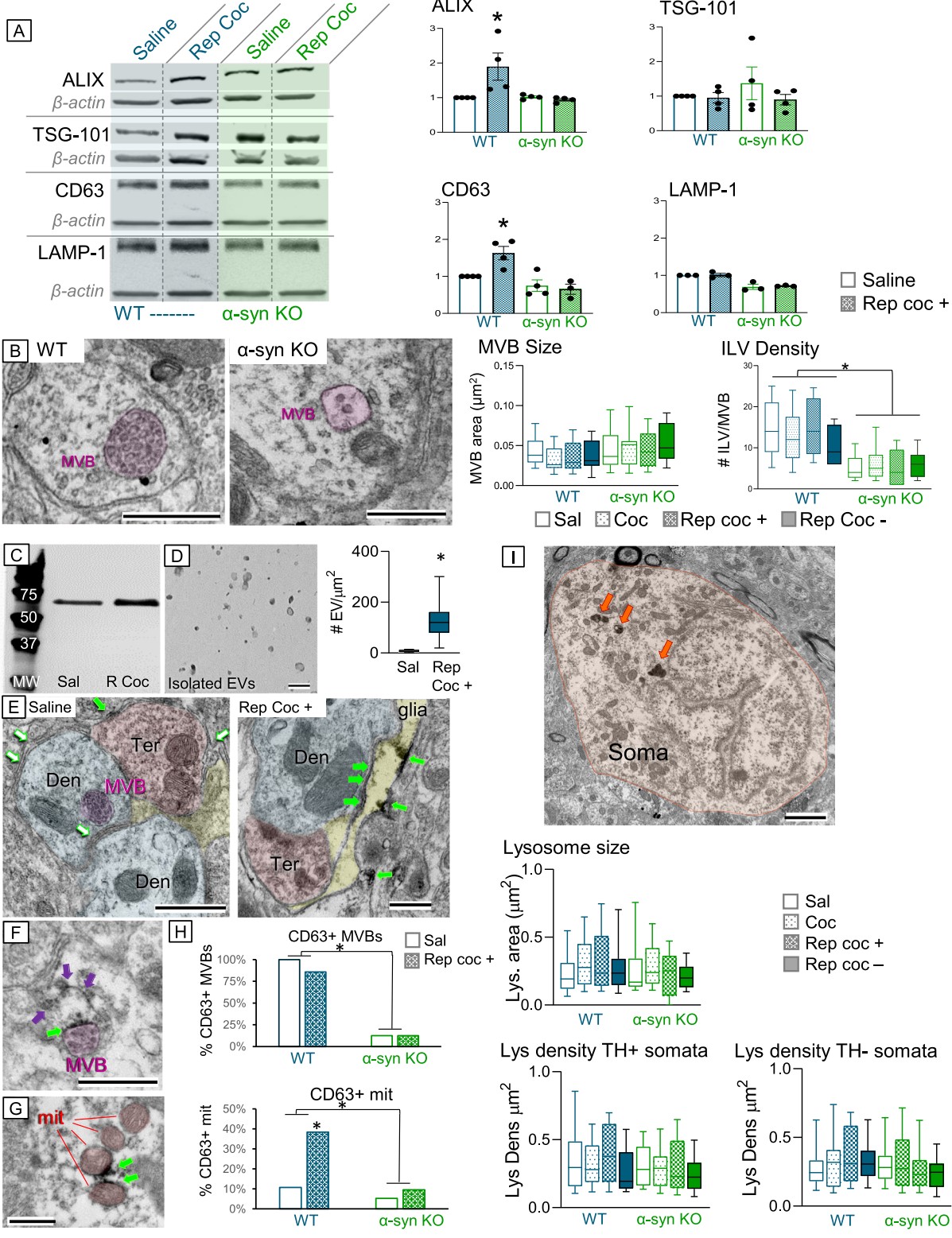

Quantification of individual lysosomal size (μm²) showed no significant change either with drug treatment ($F_{(3,329)} = 0.83$, $p = 0.48$) or by mouse genotype ($F_{(1,329)} = 1.70$, $p = 0.19$; Fig. 6i). Additional analyses showed no difference in lysosome size between different cell types in the VTA (TH+/TH−; $F_{(1,329)} = 0.23$, $p = 0.63$) or whether the cell was labeled for α-syn ($F_{(1,329)} = 1.68$, $p = 0.20$). Similarly, there were no significant differences in cell body size after cocaine administration

($F_{(3,285)} = 1.97$, $p = 0.12$), between genotypes ($F_{(1,285)} = 0.56$, $p = 0.45$), cell types (TH+/TH−; $F_{(1,285)} = 2.45$, $p = 0.12$), or whether the cell was positive for α-syn immunolabeling ($F_{(1,285)} = 1.12$, $p = 0.29$). There were also no significant differences in lysosome density either by mouse genotype (TH+: $F_{(1,203)} = 1.80$, $p = 0.18$; TH−: $F_{(1,162)} = 1.11$, $p = 0.29$; Fig. 6i) or by drug administration (TH+: $F_{(3,203)} = 1.69$, $p = 0.17$; TH−: $F_{(3,162)} = 0.40$, $p = 0.53$; Fig. 6i).

**Fig. 6 a** Western Blots for ALIX, TSG-101, CD63, and LAMP-1 in midbrain lysates obtained from WT (blue) and α-syn KO (green) and bar charts of quantification showing increased ALIX and CD63 protein levels after repeated cocaine administration that was blocked in α-syn KO mice. **b** Electron micrographs and quantification (bar chart) showing that although neither cocaine administration nor α-syn deletion affects MVB size, α-syn is critical for cargo internalization and formation of intraluminal vesicles (ILVs) within MVBs. **c** Western blot of isolated EVs from mouse brain showing increased CD63 immunoreactivity after repeated cocaine administration. **d** Electron micrograph and quantification of isolated EVs from mouse brain showing increased vesicle density after repeated cocaine. **e** Electron micrographs of VTA tissue from saline-treated mice showing CD63 immunolabeling of ILVs in a MVB (magenta outline) but virtually no labeling in the extracellular spaces (open green arrows) of saline-treated mice. In contrast, after repeated cocaine administration (left), CD63 immunoperoxidase is evident in the extracellular space (green arrows) and in glia (yellow) within the VTA. **f** Electron micrograph of a MVB (magenta) showing CD63 immunolabeling on its surface (green arrow) and vesicles with CD63 labeling (purple arrows) along a membrane that attaches the MVB to the extracellular space between adjoining cells. **g** Electron micrograph of several mitochondria (red) showing CD63 labeling (green arrows) their outer membranes. **h** Quantification of MVBs in VTA dendrites showing that α-syn KO mice have significantly less MVBs with CD63 immunolabeling than WT mice (top). Repeated cocaine increased CD63 labeling of mitochondria in the VTA, which was blocked in α-syn KO mice (bottom). **i** Electron micrograph of a non-dopamine soma showing typical lysosomal distribution (orange arrows). Quantification show that in general neither cocaine administration nor α-syn deletion affected lysosome size or density within VTA somata, although the absence of cocaine after repeated administration did significantly decrease lysosome density in both dopaminergic and non-dopaminergic somata within the VTA. MVB = multivesicular body; ILV = intraluminal vesicles; scale bar = 500 nm; *$p < 0.05$

## Discussion

The work presented here demonstrates several key findings of cocaine-mediated upregulation of α-syn and its role in substance abuse disorders. We show that α-syn is necessary for the preference for rewarding stimuli and for the cognitive flexibility needed to change previously learned behavioral strategies. Additionally, our findings indicate that the cocaine-induced increase in pre-synaptic α-syn is highly regulated and targets specific axon terminals in the VTA. This strategically positions α-syn to collectively fine-tune afferent input to VTA dopamine neurons, influencing dopamine activity. Finally, the data provide evidence that repeated cocaine administration also upregulates α-syn post-synaptically, where it is necessary for proper MVB formation and facilitates exosome release (Fig. 7).

We show that changes in α-syn modulate discrete aspects of behaviors that are affected with substance abuse. For instance, α-syn is necessary for the preference of rewarding stimuli, the cognitive flexibility needed for changing previously learned behavioral strategies, but not for locomotor sensitization to cocaine. This was initially surprising because locomotor sensitization is indicative of glutamatergic potentiation of mesolimbic dopamine neurons, a change associated with drug seeking and craving[36,37]. However, the pre-synaptic location of α-syn may explain the discrepant findings. Incentive salience (wanting/motivation) and hedonic aspects (liking/reward) of a stimulus are functionally distinct and "liking" a rewarding stimulus is not a direct action of the mesolimbic dopamine system[38]. Instead, hedonic information is mediated by afferent inputs to dopamine neurons from small regional "hotspots," which include the ventral pallidum and nucleus accumbens[39]. Increased preference for rewarding stimuli may occur by α-syn alteration of afferent signals derived from hedonic "hotspots" rather than α-syn acting directly on VTA dopamine neurons. This behavioral specificity fits with our findings that pre-synaptic α-syn upregulation by cocaine is highly regulated and is targeted to specific axon terminals in the VTA (discussed below).

Spatial memory testing shows no difference in acquisition between α-syn KO and WT mice, but α-syn KO mice show increased perseveration and have difficulty changing strategies. The data indicate that α-syn is critical for the cognitive flexibility needed for changing previously learned information (behavioral strategies), which matches early work showing that α-syn messenger RNA (mRNA) increases during developmental song acquisition in zebra finches, but dramatically decreases once song is acquired[17]. Taken together, the results indicate that cocaine-mediated α-syn upregulation is necessary for the neuronal malleability needed for substance abuse to develop by allowing changes to patterns of dopamine activity that have been previously set.

Alternately, the discrepancy between spatial learning acquisition and perseverative testing in α-syn KO and WT mice may be indicative of α-syn targeting of specific functional pathways. Spatial memory acquisition is attributed to hippocampal activity, whereas perseveration is mediated by the prefrontal cortex[40,41], a brain regions that also mediates impulsivity and cocaine craving[36,42,43].

Spatial memory testing also demonstrated a functional specificity for the type of motivation that is mediated by α-syn. The Barnes Maze protocol uses fear motivation (bright lights, white noise) to prompt the mouse to expediently find the escape hole. There were no differences in either the acquisition latency or number of errors made across trials between the α-syn KO and WT mice, demonstrating equal effects of fear motivation in these groups, which is supported by studies showing that α-syn has no significant effects on fear conditioning[44]. In contrast, when given free access, α-syn KO mice ingested less SCM than WT mice, although they preferred it to water. This demonstrates that α-syn KO mice have decreased motivation for rewarding, but not fearful stimuli, providing evidence for specific actions of α-syn in mediating the hedonic value of a stimulus.

The activity of VTA dopamine neurons is a balance of excitatory and inhibitory inputs that is disrupted with repeated drug use, causing a shift to increased glutamate activation of these cells[10,13,14,45] (Fig. 7). α-Syn expression changes in response to neuronal activity and is highest in brain regions undergoing synaptic plasticity[17,46,47]. Consistent with these ideas, we found dynamic shifts in α-syn levels within VTA axon terminals matching previously reported changes in cocaine-mediated glutamate activity in this region. Repeated cocaine administration increases α-syn in axon terminals making excitatory-type synaptic contacts onto dopamine dendrites (TH+) when it was cocaine is systemically present. Conversely, when cocaine is no longer systemically present, α-syn levels decrease in excitatory-type axon terminals, but increase in inhibitory-type axon terminals, especially those making synaptic contacts onto non-dopamine (TH−), presumably GABAergic dendrites (Fig. 4b).

The cocaine-mediated α-syn upregulation in VTA axon terminals is highly precise. α-Syn immunolabeling is not present in all axon terminals of a specific type or those making contact onto a specific dendrite (Fig. 4c). The functional shifts in α-syn levels to specific axon terminals point to a "fine-tuning" action for α-syn. By targeting specific VTA afferents, α-syn can modulate VTA dopamine activity in response to changing effects of systemic cocaine. However, it is still unclear whether α-syn acts to

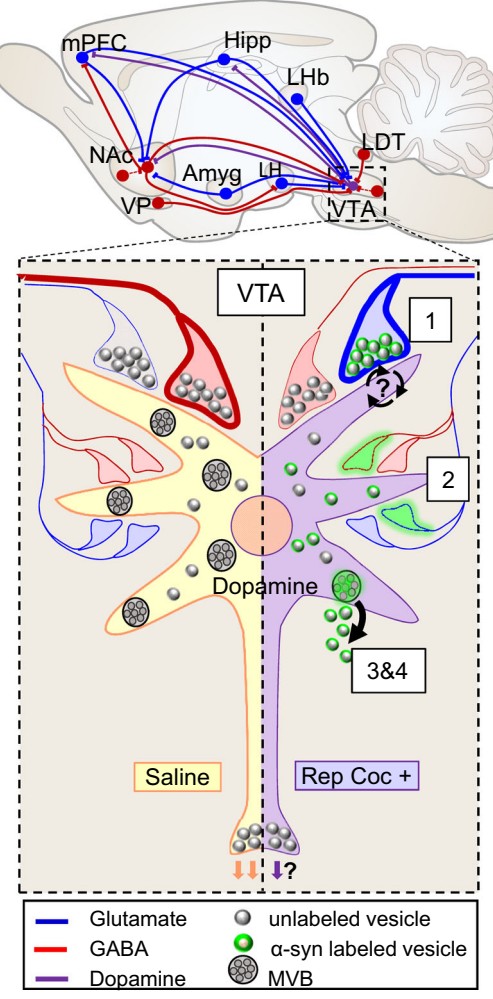

**Fig. 7** Schematic summary diagram showing the complex glutamatergic (blue) and GABAergic (red) afferent information coming to VTA dopamine dendrites (purple). Under normal conditions, dopamine neurons show steady tonic activity that is under GABAergic control (heavy red axon terminal). In contrast, repeated cocaine causes dopamine burst firing, which is under glutamatergic control (heavy blue axon). We show that (1) repeated cocaine administration increases α-syn immunolabeling (green) in glutamatergic axon terminals making synaptic contacts onto VTA dopamine neurons (purple), where it influences glutamatergic neurotransmission. However, it is still unclear whether α-syn acts to facilitate further glutamate release or is recruited to inhibit excessive glutamate activity. (2) Cocaine-mediated α-syn upregulation is dynamic and specific. We show that α-syn targets certain axon terminals, but not others making contact onto the same dendrite. a-Syn may be shifting and targeting specific axon terminals as addiction develops to modulate the input coming from rewarding/aversive brain loci, changing VTA dopamine output. (3) The cocaine-mediated decrease in MVBs is α-syn dependent and caused by increased exosome release. (4) Normal MVB formation is dependent on α-syn for cargo internalization and intraluminal vesicle insertion and abscission

facilitate or inhibit synaptic neurotransmission. Because α-syn has a primarily pre-synaptic distribution (under normal conditions), its regulatory role in synaptic neurotransmission has been extensively studied with mixed reports for both facilitatory and inhibitory actions at the synapse. A facilitating role in neurotransmission is supported by studies showing that α-syn acts as a chaperone protein during synaptic vesicle docking to promote SNARE (soluble *N*-ethylmaleimide-sensitive factor attachment protein receptor) complex assembly[20], promotes vesicle cargo

release by dilating the fusion pore[21], and that deletion of all three synuclein isoforms (α, β, and γ) inhibits long-term potentiation[47]. Inhibition of synaptic neurotransmission is supported by findings that α-syn KO mice show hyperdopaminergic function[48], increased recovery from paired-pulse depression at dopaminergic synapses[22], increased rates of dopamine vesicle pool replenishment[49], and decreased reserves of synaptic vesicle pools in hippocampal cell culture[18,19]. Moreover, when α-syn is overexpressed there is decreased vesicle fusion on chromaffin cells[23] and decreased neurotransmission in cultured hippocampal neurons[24]. The conflicting findings may indicate that α-syn has multiple functions and can act to facilitate or inhibit neurotransmission depending on the targeted cell or by changing physiological conditions of a cell induced by repeated cocaine administration.

Our findings demonstrate that α-syn is dynamic, responsive to neuronal activity, and shows highly targeted changes to specific axon terminals following repeated cocaine administration. Regardless of whether α-syn is recruited to increase or decrease neurotransmission, the selective upregulation of α-syn by cocaine to specific afferent axon terminals in the VTA appears to be a critical regulatory mechanism for changing dopamine activity and dopamine-related behaviors as addiction develops. Afferent excitatory inputs to the VTA originate largely from the medial prefrontal cortex[50]. They also come from the hippocampus, laterodorsal tegmentum, amygdala, and habenula[6,50–52], the latter two regions contributing to aversive aspects of motivated behaviors[43,53,54]. GABAergic inputs disproportionately favor the caudal VTA and arise mainly from the ventral pallidum, habenula, and laterodorsal tegmentum[6,54]. Notably, afferents from the ventral pallidum were identified as a hedonic "hotspot" mediating the likeability of rewarding stimuli[39]. These diverse inputs have been shown to target specific functional subsets of neurons in the VTA creating behavioral "microcircuits"[55]. As such, shifting α-syn upregulation to specific VTA axon terminals may influence the functional processing of rewarding and aversive inputs affecting overall dopamine output, explaining the progressively decreasing reward but increasing motivational drive/craving with chronic cocaine use.

Post-synaptic α-syn immunolabeling increased after repeated cocaine administration and was evident predominantly on endocytic vesicles, MVBs, and the outer membrane of mitochondria in dendrites and somata within the VTA. This distribution is consistent with reports of transcellular transport of α-syn via EVs[56]. However, rather than passive transport, our data indicate an active role for α-syn in vesicle internalization, MVB formation, and exosome release. Cocaine decreased the presence of MVBs in VTA dendrites and increased midbrain protein expression of ALIX, an ESCRT-I- and ESCRT-III-binding protein, and CD63, a tetraspanin found on the lipid membrane of endocytic and EVs[32,34,35]. These changes are significantly attenuated or blocked completely in α-syn KO mice, indicating that α-syn is needed for the cocaine-induced elevation in ESCRT protein activity and EV release.

The observed MVB decrease appears to be a function of increased exosome release rather than inhibition of MVB formation or increased lysosomal degradation because repeated cocaine increased exosome densities in midbrain tissue samples and whole-brain exosome isolation preparations and increased CD63 immunolabeling in the VTA (Fig. 6). Cocaine-naive mice had extremely sparse CD63 labeling, which was limited to ILVs within MVBs. After cocaine, CD63 labeling was prevalent between cells within the VTA demonstrating the release of vesicles from MVBs into the extracellular space. In contrast, cocaine administration did not alter MVB size or ILV density as compared to saline controls, confirming that MVBs developed

similarly in the presence or absence of cocaine. Moreover, cocaine did not significantly alter LAMP-1 protein expression from midbrain tissue samples or VTA lysosome size. In fact, there is a decrease in lysosomal density in VTA somata after repeated administration when cocaine is no longer systemically active (Fig. 6i). The increased exosome release may serve to change the activity and/or signaling of neighboring cells because exosomes can transfer and affect different cell types including glial cells[57]. They contain numerous cytoplasmic proteins, including receptors[58], regulatory lipids[59], and mRNA/microRNA, cargoes that are biologically functional and changes depending on the physiological state of the releasing cell[60,61]. As such, the increase in exosome release witnessed with repeated cocaine administration may provide an additional mechanism for changes in midbrain dopamine function associated with the development of substance abuse.

The depletion of MVBs and increased exosome release induced by repeated cocaine administration was attenuated in α-syn KO mice, indicating that α-syn critically contributes to these actions. α-syn is a promiscuous protein that binds to highly curved lipids, including endocytic vesicles via its C-terminus[62]. It facilitates synaptic vesicle release through its promotion of SNARE complex assembly[20] and its overexpression results in vesicle release from MVBs in vitro[63–65]. SNARE proteins are also present on the MVB membrane that act to release ILVs[64,65] pointing to parallel mechanisms for vesicle release by α-syn pre- and post-synaptically.

Because cocaine-mediated ALIX elevation was also blocked in α-syn KO mice, we examined MVB size and ILV density between the two genotypes. ALIX works with cellular ESCRT machinery[30–33] and independently to initiate cargo internalization and vesicle abscission during MVB formation. Repeated cocaine administration has no effect on either MVB size or ILV density (~10–20 vesicles/MVB). While α-syn deletion has no significant effect on MVB size, it decreases the number of vesicles observed within each MVB (~5–8 vesicles/MVB; Fig. 6b). In addition, α-syn KO mice have many large vacuoles that are completely devoid of vesicles, indicative of increased numbers of early endosomes, which are sparse in WT mice. These findings indicate that α-syn is critical for ALIX-mediated cargo/vesicle internalization and normal MVB development.

The data presented here show that α-syn modulates specific aspects of addictive behavior, particularly hedonic information and motivation to pursue rewarding stimuli, by targeted presynaptic modulation of the diverse afferent input onto VTA dopamine neurons. Moreover, we show that α-syn upregulation in regions undergoing glutamatergic plasticity is vital for neuronal malleability necessary for changing set behavioral patterns. We also show that chronic cocaine exposure increases postsynaptic α-syn where it is necessary for normal cargo/vesicle formation and exosome release, pointing to similar functions for α-syn pre- and post-synaptically.

## Methods

**Subjects.** A total of 132 adult (14+ weeks) male WT (C57BL66/J) and α-syn KO (C57BL/6N-Snca$^{tm1Mjff}$/J) mice were utilized for experimental procedures (Jackson Labs, Bar Harbor, ME). C57BL/6N-Snca$^{tm1Mjff}$/J mice have a targeted mutation of exons 1–4 of the α-synuclein gene effectively disrupting the α-synuclein gene. All mice were group housed (2–4 mice per cage) in a temperature- and humidity-controlled facility on a 12 h light/dark cycle with food and water available ad libitum. Experimental protocols were approved by the Institutional Animal Care and Use committee at Weill Cornell Medical College and performed in accordance with the Guidelines for the Care and Use of Laboratory Animals of the National Institutes of Health and animal procedures were outlined following ARRIVE guidelines[66].

**Statistics and reproducibility.** The exact sample size and descriptions of statistical analyses are reported for each individual experiment. Measures for each study were derived from distinct samples, except for behavioral studies where repeated test trials were conducted and the brain tissues from these mice were processed for EM analysis. All data generated or analyzed during this study are included in this published article and its Supplementary Information files.

**Drug administration.** Both WT and α-syn KO mice were randomly assigned to one of four treatment groups: a single cocaine injection (coc); repeated cocaine injections with cocaine systemically present at time of tissue preparation (rep coc+); repeated cocaine injections with cocaine systemically absent at time of tissue preparation (rep coc−); and saline control (sal). Mice received a single intraperitoneal (i.p.) injection of cocaine hydrochloride (15 mg/kg) mixed fresh daily in sterile saline per day for either 1 day (coc) or 7 consecutive days (rep coc; Sigma-Aldrich, St. Louis MO). For the coc and rep coc+ experimental groups, tissue for immunoblotting and microscopy was processed within 15 min of the last cocaine injection, a time point at which we previously reported detectable levels of cocaine and its metabolite, benzoylecgonine, in the blood[14]. For the rep coc− group, tissue was processed 72 h after the last drug injection, a time point with no discernable systemic levels of cocaine or cocaine metabolites[2]. Mice in the rep coc− group were assessed and showed little to no somatic withdrawal symptoms immediately prior to tissue processing (Supplementary Table 1).

**Behavioral measures.** *Open field/locomotor activity.* Locomotor activity was measured using automated open field chambers (Med Associates, Fairfax, VT). WT (n = 12) and α-syn KO (n = 12) mice were acclimated to the chambers for one day prior to the start of testing. After acclimatization, mice were assessed on locomotor activity for 30 min, once daily for 8 consecutive days. On day 1, baseline locomotor activity was measured for all mice. On days 2–8 mice received systemic injections of either cocaine (15 mg/kg, i.p.) or saline immediately prior to placement into the open field where locomotor activity was recorded. Mice were returned to their home cages after testing. A two-factor (drug $\times \times$ genotype) repeated-measures, mixed-design analysis of variance (ANOVA) was used to analyze locomotor activity data utilizing the Tukey's test for post hoc group analysis.

*Place preference conditioning.* Preference to cocaine was measured using automated CPP boxes that have two distinct sides, both in spatial and tactile cues (Med Associates, Fairfax, VT). WT (n = 12) and α-syn KO (n = 12) mice were acclimated to the test chambers for 30 min one day prior to the start of testing. On day 1 of testing, mice were placed into CPP boxes for 30 min with no drug administration and access to both sides of the box. The time spent in each chamber was recorded to ensure there was no side preference. Mice that displayed a side preference (57% or greater time spent in a specific side) were excluded from the study. On days 2–8, mice received two conditioning trials per day (10:00 a.m. and 6:00 p.m.). During the conditioning trials, cocaine (15 mg/kg, i.p.) was paired with one side of the CPP box and saline (equal volume, i.p.) was paired with the other side. Mice were placed into the corresponding box side immediately after systemic drug injection with the access doors closed. The order of drug administration, time of day, and chamber side were counterbalanced to prevent bias. On day 9, no drugs were injected and mice were placed in the CPP boxes (12:00 p.m.) with the access doors open. The time spent in each chamber was recorded to assess preference to repeated cocaine administration. A two-factor (drug × genotype) ANOVA was used to analyze the CPP data and a Tukey's test was used for post hoc group analysis.

*Barnes maze.* Spatial memory was assessed using a Barnes maze, an elevated circular platform (36 in. diameter) with 20 round holes (2 in. diameter) around the circumference of the maze. WT (n = 8) and α-syn KO (n = 8) mice were given two trials per day for 4 consecutive days to find the escape box using spatial cues. Each trial lasted 5 min. Mice were placed in the center of the platform in a start box. The trial began as soon as the box was lifted. Mice were free to explore the platform and find the escape hole, one of the 20 holes of the platform that had an escape box attached to the underside of the platform so that it was not visible to the mouse. The escape box was consistently placed at the "6 o'clock" position of the platform (Fig. 2b). If at the end of the first trial the mouse did not enter the escape box, the mouse was gently guided to the escape box to be made aware of its existence. This was only done on the first trial. The latency to enter the escape box and the number of errors was measured for each mouse. An error consists of any entry attempt into an incorrect hole. Locomotion, latency, errors, and mouse positioning was automatically monitored and scored via real-time processing of video recording by the Any-Maze program (Stoelting, Wood Dale, IL). A repeated-measures ANOVA was used to examine differences between WT and α-syn KO mice in latency to find the escape hole and errors made across learning trials.

*Perseveration testing.* On day 5, the position of the escape box was moved to assess the effects of prior learning on the ability to learn a new escape position. The new escape box position was set approximately at the "11 o'clock position" (Fig. 2b), and mice were tested for three 5-min trials. The latency to enter the escape box, the number of errors (wrong entry attempts), perseverative errors (entries into original escape box position), and the time spent in the original escape box position quadrant were monitored and scored via real-time processing of video recording by the Any-Maze program (Stoelting, Wood Dale, IL). A two-factor (drug × genotype) repeated-measures ANOVA was used to assess changes in the latency to find the escape hole, number of errors, perseverative errors, and perseverative time across test trials.

*SCM CPP and intake.* Preference to SCM followed a similar protocol to cocaine CPP testing; however, mice received only one conditioning trial per day (11:00 a. m.), rather than two. WT ($n = 6$) and α-syn KO ($n = 6$) mice were acclimated to the CPP test box for 1 day. On test day 1, mice were placed in the CPP box with access doors opened to test for side preference. Mice that displayed a side preference were excluded from the study. On days 2–8, mice were placed in the CPP box for 30 min with access to both sides of the box. One side was supplied with water, while the other side was supplied with SCM in a small dish. Water and SCM intake was measured daily for each mouse and the chamber side pairing for water/SCM was counterbalanced. On day 9, no water or SCM milk was provided. Mice were placed in the test chamber with access to both sides for 30 min and the amount of time in each side was measured. Differences in water and SCM intake across trials were analyzed using a two-factor (solution × genotype) repeated-measures, mixed-design ANOVA. A two-factor (solution × genotype) ANOVA was used to analyze the CPP data.

*Withdrawal and anxiety.* The somatic aspects of withdrawal were assessed 72 h after the last cocaine injection (rep coc−) in both WT ($n = 6$) and α-syn KO ($n = 6$) mice using a 10-point symptom severity scale and assessing the presence of 16 physiological traits of withdrawal. To test the potential emotional aspects of withdrawal, WT ($n = 12$) and α-syn KO ($n = 12$) mice that were naive to cocaine (sal) or in withdrawal (rep coc−) were tested on a battery of anxiety measures, including: the elevated zero maze; assessed on their position (periphery vs. center) in an open field apparatus for a 30 min test period; and measured the rate of defecation in a 30 min test period. WT mice showed an increase in anxiety-like behaviors 72 h after repeated cocaine administration as compared to saline controls (Supplementary Fig. 2). These mice spent less time in the open areas of the zero maze ($F(1,11) = 8.26$, $p = 0.017$) and made fewer attempts to enter the open areas ($F(1, 11) = 9.06$, $p = 0.013$; Supplementary Fig. 2). In addition, they had an increased number of fecal pellets emitted in a 30-min test period ($F(1,11) = 28.21$, $p = 0.001$). However, there were no significant differences in the duration of time these mice spent in the periphery (as compared to the center) of the open field apparatus ($F(1,11) = 0.10$, $p = 0.76$; Supplementary Fig. 2).

It was more difficult to determine anxiolytic effects of cocaine withdrawal in the α-syn KO mice because these mice displayed higher levels of anxiety in general as seen in the open field where α-syn KO mice, regardless of cocaine exposure, show increased time in the periphery of the open field as compared to WT mice (Supplementary Fig. 2B; $F(1,23) = 6.26$, $p = 0.021$). However, there is some evidence that cocaine withdrawal further increases anxiety in α-syn KO mice because these mice spend less time in the open areas of the zero maze ($t(1,10) = 3.14$, $p = 0.01$) and have greater rates of defecation ($t(1,10) = 2.74$, $p = 0.02$) as compared to cocaine-naive α-syn KO mice. There were no significant differences in ambulatory speed either by cocaine exposure ($F(1,23) = 0.0009$, $p = 0.934$), but the α-syn KO mice were generally slower than their WT counterparts ($F(1,23) = 8.45$, $p = 0.009$).

**Immunoblotting**. *Tissue preparation for Western blots.* Repeated cocaine-treated and saline control WT and α-syn KO mice ($n = 16$; 4/group; 4 groups total) were anesthetized with isoflurane (4%, 0.7 l/min $O_2$) and decapitated for rapid removal of the brain 15 min after the last drug injection. The tissue was immediately placed on a chilled Petri dish and a 2.5 mm$^3$ portion of midbrain containing the VTA was isolated (~−1.70 to −4.20 Bregma), then flash frozen with liquid nitrogen, and stored at −80 °C so that tissue from all experimental groups could be processed together.

*Western blots.* Tissue samples were homogenized in RIPA (50 mM Tris-HCl, pH 7.4, 150 mM NaCl, 1.0% (v/v) NP-40, 0.5% (w/v) sodium deoxycholate, 1.0 mM EDTA, 0.1 (v/w) sodium dodecyl sulfate (SDS)) buffer including 1% protease inhibitors (Roche, Mannheim, Germany) using a 2 ml Teflon tissue homogenizer. Equal amounts of protein (25 μg) were separated by SDS-polyacrylamide gel electrophoresis (SDS-PAGE) and transferred to polyvinylidene fluoride (PVDF) membranes. Antibodies used were anti-α-syn (1:2000, BD Biosciences, #610787), anti-CD63 (1:2000, Millipore Sigma, #SAB4301607), anti-ALIX (1:2000, Thermo Fisher Scientific, #MA1-83977), anti-LAMP-1 (1:1000, Abcam, #AB208943), anti-TSG-101 (1:2000, Thermo Fisher Scientific, #MA1-23296), and anti-β-actin (1:3000, Cell Signaling Technology, #3700 or #4970). Protein bands were detected using a LI-COR digital imaging system and band intensities were quantified with the Image Studio software. Differences between experimental groups were analyzed using a two-factor (drug × genotype) ANOVA.

**Immunocytochemistry and microscopy**. *Tissue preparation for microscopy.* WT and α-syn KO mice (32 mice; $n = 4$/group) were deeply anesthetized with sodium pentobarbital (300 mg/kg, i.p., Sigma) and perfused through the heart with 5 ml of heparin in saline (1000 U/ml; American Pharmaceutical Partners). This was followed by 30 ml of 3.75% acrolein and 2% paraformaldehyde (PFA) mix in 0.1 M phosphate buffer (PB; Polysciences) and an additional 50 ml of 2% PFA solution only (Sigma). For confocal studies, the fixative perfusion procedure was similar except that acrolein was omitted and a 4% PFA concentration was used. After perfusion, the brain was removed from the cranium, cut into 3–4 mm coronal blocks, and post fixed in PFA for 30 min. The tissue was then cut into 40-μm sections using a Leica Vibratome (VT1000). The collected sections were placed into

storage solution (30% sucrose, 30% ethylene glycol in 0.05 M PB (pH 7.4)) and stored at −20 °C until use for immunolabeling/microscopy.

*Tissue selection.* Coronal sections of tissue containing the VTA were processed for confocal and EM experiments. Tissues from all experimental groups were co-processed to eliminate immunolabeling variability bias. The medial/caudal region of the VTA (−3.30 to −3.50 Bregma) was selected based on the greatest density of TH+ immunoreactive cells and the presence of both the paranigral and parabrachial sub-regions in the section. TH is the rate-limiting enzyme for catecholamine production and in the VTA is used for identification of dopamine neurons. Precise regional selection of sections for the studies was important to prevent potential bias from known differential distributions of dopamine and GABA neurons through the rostral–caudal axis of the VTA[67]. Antibody controls included staining KO tissue and omitting the primary antibody.

*Confocal microscopy.* The tissue sections were rinsed in 0.05 M phosphate-buffered saline (PBS; pH 7.4) and then blocked in 0.25% triton, 0.5% bovine serum albumin (BSA), and 5% normal donkey serum (NDS) in 0.05 M PBS for 2 h. After PBS rinse, the sections were incubated overnight in primary antibodies prepared in 0.1% Triton and 0.5% BSA in 0.05 M PBS. The primary antibodies used for the triple-labeled study included a rabbit polyclonal anti-glutamate antibody (1:500 dilution (dil); Millipore Sigma AB133), a rat polyclonal anti-GABA antibody (1:500 dil; Thermo Fisher Scientific; AB175), and a mouse monoclonal anti-α-syn antibody (1:1000 dil; BD Biosciences; #610787). The following day, the sections were rinsed using PBS and incubated in the secondary antibodies mixed in 0.1% Triton and 5% NDS in 0.05 M PBS for 2 h. Secondary fluorescent antibodies included donkey anti-rabbit Cy3 (#711-165-152), donkey anti-rat Alexa Fluor 647 (#712-605-153), and donkey anti-mouse FITC (#715-095-150) all utilized at a 1:400 dilution and obtained from Jackson Laboratories. After incubation, the sections were rinsed, mounted on slides with Vectashield (Vector Laboratories), and imaged with an SP5 confocal microscope equipped with HeNe/Ar lasers (488, 543, and 633 nm; Leica SP5). Image stacks were combined using the Image J software.

Labeled puncta were counted from a single field and categorized by spectrometry for single or dual labeling of antibodies. Counts only included puncta labeled for: (a) glutamate, (b) GABA, (c) dual labeled with α-syn/glutamate, or (d) dual α-syn/GABA to examine changes in subcellular α-syn localization within these cell types. However, there were VTA puncta that contained α-syn immunolabeling alone that were not included in the counts. The VTA is a heterogeneous structure and these puncta most likely include axon processes containing other neurotransmitters or neuromodulatory peptides. $\chi^2$ analysis was used to assess group differences in labeled puncta.

*Electron microscopy.* Selected coronal sections of tissue obtained from WT and α-syn KO mice ($n = 32$ (4 mice per group/8 experimental groups) were processed simultaneously to prevent bias. Sections were placed in 1% sodium borohydride in 0.1 M PB to remove excess aldehydes, rinsed, and then incubated in 0.5% BSA in 0.1 M Tris-buffered saline (TBS) to minimize non-specific labeling. After BSA incubation, sections were rinsed and incubated for 48 h at room temperature in a primary antibody solution in 0.1% BSA in TBS. EMc studies utilized a monoclonal mouse anti-α-syn antibody (1:1000 dil; BD Biosciences; #610787), a polyclonal rabbit anti-CD63 antibody (1:1000 dil; SBI systems; EXOAB-CD63A-1), and a polyclonal sheep anti-TH antibody (1:10,000 dil; Millipore Sigma; AB1542). After primary antibody incubation, tissue was rinsed with 0.1 M TBS to prepare for secondary antibody incubation. Peroxidase labeling of α-syn was achieved using a biotinylated horse-anti-mouse immunoglobulin G (IgG) (1:400 dil; Vector Labs; BA-2001) and of CD63 using a biotinylated donkey anti-rabbit IgG (1:400 dil; Jackson Laboratories; #711-065-152) in 0.1% BSA in 0.1 M TBS for 30 min, rinsed, and then placed in an avidin–biotin solution (1:200 dilution in 0.1 M TBS; Vector Laboratories) for 30 min. The peroxidase reaction product was visualized with 0.022% 3,3′-diaminobenzidine (Aldrich) and 0.003% $H_2O_2$ in 0.1 M TBS for 6 min. For immunogold-silver visualization of TH immunoreactivity, tissue sections were incubated for 1 h in donkey anti-sheep colloidal gold (1 nm) IgG (1:50 dilution; EMS; #25820), fixed in 2% glutaraldehyde, and enhanced with a silver solution (IntenS-EM kit: EMS) for 7 min. The tissue was post fixed with 2% osmium tetroxide in 0.2 M PB for 1 h, dehydrated through a series of ethanols (30, 50, 70, 95, and 100%), propylene oxide, and then a 50:50 mixture of propylene oxide and Embed 812 Epon substitute (EMS) overnight. The following day, the sections were incubated in the Epon substitute for 2 h, embedded between two sheets of Aclar plastic, and placed in an oven (60 °C) for 24 h. The VTA was isolated from the embedded tissue and cut to 70 nm using a diamond knife and ultratome (Leica). Sections were collected onto 400 mesh copper grids (EMS) and imaged using a transmission electron microscope (Technai 12 Biotwin or Phillips CM-10).

*Ultrastructural analysis.* The ultrastructural analysis of the tissue was conducted at ×5800–10,500 magnification (for inclusion of somata) and ×25,000–46,000 magnification (for axonal and dendritic profiles and analysis of intracellular organelles). For each specific ultrastructural analysis, care was taken to ensure equal sampling from experimental groups. For pre- and post-synaptic evaluation of α-syn distributions in WT mice, a total of 15,870 μm$^2$ in 820 micrographs were analyzed (sal: 211 micrographs/4194 μm$^2$; coc: 195 micrographs/3883 μm$^2$; rep coc+: 198 micrographs/3904 μm$^2$; rep coc−: 216 micrographs/3889 μm$^2$). For morphological analysis of MVBs, lysosomes, and vesicular density, a total of 43,314 μm$^2$ in 2182 micrographs were analyzed (WT sal: 313 micrographs/5799 μm$^2$; WT coc: 292 micrographs/5786 μm$^2$; WT rep coc+: 288 micrographs/5694 μm$^2$; WT rep coc−:

294 micrographs/5823 µm²; α-syn KO sal: 245 micrographs/4898 µm²; α-syn KO coc: 233 micrographs/5056 µm²; α-syn KO rep coc+: 262 micrographs/5208 µm²; α-syn KO rep coc−: 255 micrographs/5050 µm²). For subcellular CD63 distributions in WT and α-syn KO mice, a total of 13,513 µm² in 680 micrographs were analyzed (WT sal: 162 micrographs/3226 µm²; WT rep coc+: 170 micrographs/3381 µm²; α-syn sal: 177 micrographs/3502 µm²; α-syn rep coc+: 171 micrographs/3404 µm²).

The neuronal and glial elements in the tissue were identified using criteria of Peters et al.[68]. The analysis of the tissue was consistently confined to the outmost surface of tissue sections in order to overcome limited penetration of the immunoreagents. To verify that tissue used for analysis was selected at similar levels, a within-group ANOVA was completed on TH immunogold labeling and found no significant differences in immunogold density between individual animals within each treatment group ($F$ (14,1189) = 1.164, $p > 0.05$).

The parameters used for between-group comparisons include changes in: subcellular α-syn distribution based on identified cell type and morphological synaptic contact, MVB quantity and vesicle content, lysosome density and size within identified cell bodies, and CD63 subcellular distributions. Data were analyzed using $\chi^2$ analysis for proportional changes of α-syn location and two-factor (drug × neuronal regions) ANOVA to detect changes in regional α-syn distribution within VTA neurons. For changes in lysosomal morphometry, data were analyzed using a three-factor (drug × genotype × α-syn immunolabeling) ANOVA of TH+ and TH− somata of lysosome size and density within specified cell bodies in the VTA.

**EV isolation, array blot, and subcellular quantification.** *EV antibody array blot.* EVs isolated from blood samples obtained from WT saline- and cocaine-treated mice were assessed for known associated proteins using an Exosome Antibody Array Blot (System Biosciences) to identify potential targets for cocaine-induced changes. EVs were isolated from blood samples obtained from WT mice (*n* = 9; 3/group) that received repeated cocaine (15 mg/kg, i.p.) or equal volume saline injections for 7 consecutive days. Either 15 min or 72 h after the last injection, mice were deeply anesthetized with sodium pentobarbital (300 mg/kg, i.p.), the chest opened, and 1 ml of blood was obtained by injecting a syringe with a 22-gauge needle into the left ventricle of the heart. The plasma was separated via centrifugation and processed using a kit to obtain pure exosome samples (ExoQuick Ultra, SBI). The isolated exosomes were resuspended in PBS and protein quantification was determined using Qubit assay. A 50 µg sample from each mouse was processed using lysis and labeling buffers. Excess labeling reagent was removed and the labeled EV lysate was eluted by centrifugation (800 × *g*; 2 min). Five milliliters of labeled EV lysate was pipetted on the array membrane and incubated overnight at 4 ℃. The following day, the membranes were washed and developed. Signal intensities were quantified relative to positive controls with Image Studio software. ANOVA was used to analyze differences between experimental groups. There was no evidence of cellular contamination as determined by the absence of GM130 (*cis*-Golgi matrix protein) reactivity on the blot (Fig. 6d).

*EV immunoblot and EM analysis.* EVs were isolated from whole-brain samples obtain from WT mice (*n* = 12; 6/group) that received repeated cocaine (15 mg/kg, i.p.) or equal volume saline injections for 7 consecutive days. Mice were anesthetized with isoflurane (4% in 0.7 l/min O₂), decapitated, and the brain was quickly removed and placed immediately into warmed Hibernate E solution (37 °C) with papain (20 U/ml) to gently homogenize the tissue without breaking apart. The solution was centrifuged (3000 × *g* for 15 min) and then progressively vacuum filtered (40–0.2 µm) to remove cellular debris. The remaining supernatant was processed by a commercially available kit for EV isolation (ExoQuick-TC, SBI). Half the samples were used for immunoblotting and the other half was used for EM analysis.

*Immunoblotting.* The EV pellet was resuspended in RIPA (50 mM Tris-HCl, pH 7.4, 150 mM NaCl, 1.0% (v/v) NP-40, 0.5% (w/v) sodium deoxycholate, 1.0 mM EDTA, 0.1 (v/w) SDS) buffer including 1% protease inhibitors (Roche, Mannheim, Germany). Equal amounts of protein (10 µg) were separated by SDS-PAGE and transferred to PVDF membranes. EV quantities were evaluated using a polyclonal rabbit anti-CD63 antibody (1:1000 dil; SBI systems; EXOAB-CD63A-1). CD63 is a tetraspanin found on the lipid membrane of EVs. Protein bands were detected using a LI-COR digital imaging system and band intensities were quantified with the Image Studio software. Differences between experimental groups were analyzed using a *t* test.

*EM isolated EVs.* For microscopic visualization and quantification, the isolated EV pellet was resuspended in 50 µl of 2% PFA in 0.1 M PB and 5 µl of solution was placed on a Formvar carbon-coated copper grid (EMS). After a 20 min drying period, the grids were rinsed with 0.1 M PBS and post fixed with 1% glutaraldehyde. The grids were then rinsed with distilled water and counterstained with uranyl acetate (EMS) for visualization. The samples were imaged using a transmission electron microscope (Technai 12 Biotwin) and EV densities were measured and differences between experimental groups were assessed using an independent *t* test.

**Reporting summary.** Further information on research design is available in the Nature Research Reporting Summary linked to this article.

## Data availability
The data that support the findings of this study are available from the corresponding author upon reasonable request. Original blot images corresponding to those shown in the main figures are available as Supplementary Fig. 3.

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

## Acknowledgements

We gratefully acknowledge funding from the Feil Family Brain and Mind Research Institute and NIH funding NS067078 to P.Z. We thank Virginia Pickel and Teresa Milner for editing and Nyi-Rein Kyaw for assistance with the Barnes Maze.

## Author contributions

O.T. contributed to manuscript writing, EM of CD63, and analyzing/quantifying electron microscopic data; A.E.L. completed the behavioral studies and MCID analysis of micrographs; L.Q. and P.Z. completed the immunoblot studies; D.A.L. conducted behavior, completed all exosome/EV isolation procedures, completed confocal and EM analysis of α-synuclein, synaptic quantification, and lysosomal morphometry, all statistical analyses, and manuscript writing.

## Competing interests

The authors declare no competing interests.
