## [Peer Review File · Communications Biology]

Reviewers' comments:

Reviewer #2 (Remarks to the Author):

The manuscript from Trubetckaia et al. outlines the effects of alpha-synuclein (a-syn) on VTA circuitry and cocaine-associated behaviors. Specifically, this work uses a KO mouse model to assess the role of a-syn on various behavioral tasks and synaptic remodeling in the VTA in response to cocaine. First, the authors demonstrate that a-syn KO animals have impaired cocaine-induced CPP and reversal learning in the Barnes Task, but spared cocaine sensitization and CPP induced by a natural reward (sweetened condensed milk, SCM). Lastly, using electron microscopy, authors demonstrate altered VTA synaptic connectivity that is potentially mediated by a-syn-mediated protein expression. While this work is of great interest to the readership and provides a potential role for a-syn independent of Parkinson's Disease, enthusiasm is tempered by a lack of a causal link between a-syn and the downstream effects on connectivity and behavior to effectively support the conclusions proposed. Moreover, limited sample size in a subset of experiments limits the interpretability of the effects. My concerns are listed below as Major and Minor:

Major

1. Although the electron microscopy experiments are incredibly elegant (and understandably costly), the sample size should be increased to 2-3 in order to draw reliable conclusions from these studies. Moreover, key groups are missing from a subset of experiments. For example, experiments presented in Figs. 5 & 6 lack a-syn-KO animals treated with acute cocaine (while there is WT coc) and lacks a-syn-KO animals without repeated cocaine on board (while there is WT coc -). These groups would provide a more representative analysis of a-syn-dependent responses to cocaine treatment.

2. While the use of both cocaine and a natural reward (sweetened condensed milk) strengthens the impact of these studies, there are missing key data that could alter the interpretation of the presented experiments. A-syn KO animals show blunted sweetened condensed milk consumption at baseline compared to WT controls (Fig 2C, Day 1). This decreased baseline consumption confounds the interpretation of SCM CPP reported in Fig 2D, as it is possible differing levels of SCM consumption during conditioning sessions is masking an effect on SCM CPP. Total SCM consumed during conditioning sessions should be reported and it would be interesting to see if SCM consumption correlates with individual CPP scores (and if this is different in WT vs KO animals).

3. While the effects on the Barnes Task are interesting, the authors fail to address how this is linked to a-syn mediated VTA function. As the underlying hypothesis is that a-syn in the VTA is critical for behavioral strategies used in these tasks, demonstration that Barnes Maze performance/training alters endogenous a-syn function (while a full electron microscopy assay of vesicle formation and connectivity is unnecessary, authors should include at minimum effects of Barnes Maze reversal learning on a-syn expression, and subsequent changes in ALIX and CDH6 expression) to draw these conclusions.

4. A major concern in constitutive knockout animal models is differentiating the role of alpha-synuclein in neural development versus its role in the adult brain, identifying a brain-region specific effect of a-syn, and generating a subsequent causal relationship. For example, in Fig 5B left, saline-treated a-syn KO animals show baseline differences in MVB in TH+ dendrites compared to WT controls. Moreover, as both the Barnes Maze and CPP tasks are also dependent on the hippocampus, it is possible that the identified effects on behavioral performance are being mediated independently of the VTA (or at least being mediated by hippocampal glutamatergic projections onto TH+VTA neurons, as is suggested by Fig. 4). Therefore, it is difficult to draw a causal relationship between alpha-synuclein function and alterations in VTA connectivity/molecular response to cocaine. This can be addressed in two ways:

- a. Does over-expression of a-syn in the VTA of a-syn KO animals restore behavioral performance and cocaine-induced effects on ALIX and CD63 ?
- b. Does deleting a-syn in the VTA of adult WT animals induce the same response pattern to cocaine as seen in the a-syn KO animals?

Minor

1. The author's claim that they have demonstrated an interaction between alpha-synuclein and ALIX is inaccurate. Their data demonstrates a potential relationship with ALIX, however, to demonstrate a direct interaction between these 2 proteins, the authors will need to provide data from a co-immunoprecipitation, FRET, or Yeast-2-Hybrid experiment. Thus, the language in the abstract and discussion should be changed to address this.
2. Total locomotor data for CPP experiments should be reported.
3. Throughout the manuscript, the authors reference CPP results as a measure for motivation. However, the conditioned place preference task measures associative memory for rewarding (or aversive stimuli), not motivation. This should be addressed throughout the manuscript and discussion.
4. The dose of cocaine used for CPP experiments should be noted in the methods section.
5. The results section often references data without an appropriate reference for the corresponding figure or statistics and should be re-written for clarity.

Reviewer #3 (Remarks to the Author):

The manuscript by Trubetckaia et al. outlines the role of α -synuclein in cocaine behaviors, cocaine changes in presynaptic α -syn distribution and expression, and cocaine-induced postsynaptic changes in exosome release and MVB formation (dependent on α -syn). Each experiment was carefully performed and individually clearly presented. There are a couple of specific comments that pertain to the overall theme or presentation of the manuscript.

1. The authors do an excellent job of outlining specific changes that occur in α -syn expression within the VTA following cocaine exposure. However, it seems the relationship between behavior expressed in Figure 1 (cocaine CPP) / Figure 2 (Barnes Maze) and their in depth analysis of α -syn changes in the VTA following cocaine is not entirely clear. There are attempts to explain the relationship throughout the discussion, but without direct manipulations of VTA α -syn in cocaine treated animals, or at least correlations between VTA α -syn change and changes in behavior, it is difficult to conclude that the behavior is related to these well-outlined changes in α -syn. Wouldn't α -syn KO be through the whole brain in KO mice, and α -syn expression increase through the whole brain in cocaine-treated animals? The discussion pertaining to not knowing what α -syn is doing to release of Glutamate, GABA, and dopamine within the VTA reinforces this notion.
2. Perhaps related to comment #1, this manuscript present a number of findings, each of which would likely stand on its own with respect to independent publications, but together form a hodgepodge of stories that is difficult to tie together (and ultimately link to behavior as in comment #1). How do pre- and post-synaptic changes tie in to the behavior.
3. 72 hours post cocaine was chosen as a time that is presented not as a withdrawal time, but a time for cocaine to no longer be present. Care was taken to show little to no somatic withdrawal signs. There was no mention (one way or the other) or control for psychological signs of withdrawal that may precede somatic signs at this time point (e.g., anxiety-like behavior)?

Minor.

1. Figure 4B labels. Asyn is used instead of α -syn (greek letter) like other figures.

Suggested Changes from Reviewer #2

1. *Although the electron microscopy experiments are incredibly elegant (and understandably costly), the sample size should be increased to 2-3 in order to draw reliable conclusions from these studies. Moreover, key groups are missing from a subset of experiments. For example, experiments presented in Figs. 5 & 6 lack asyn-KO animals treated with acute cocaine (while there is WT coc) and lacks asyn-KO animals without repeated cocaine on board (while there is WT coc -). These groups would provide a more representative analysis of a-syn-dependent responses to cocaine treatment.*

- A. All electron microscopic studies utilized **four** mice per experimental group (see above).
- B. The alpha-synuclein KO coc and rep coc – groups have been added (see above).

2. *While the use of both cocaine and a natural reward (sweetened condensed milk) strengthens the impact of these studies, there are missing key data that could alter the interpretation of the presented experiments. A-syn KO animals show blunted sweetened condensed milk consumption at baseline compared to WT controls (Fig 2C, Day 1). This decreased baseline consumption confounds the interpretation of SCM CPP reported in Fig 2D, as it is possible differing levels of SCM consumption during conditioning sessions is masking an effect on SCM CPP. Total SCM consumed during conditioning sessions should be reported and it would be interesting to see if SCM consumption correlates with individual CPP scores (and if this is different in WT vs KO animals).*

—

The requested information for the total SCM consumption during conditional sessions was already reported in Figure 2C (daily intake in ml) in the original manuscript. However, we greatly appreciate the reviewer's recommendation for a correlational analysis between SCM intake and side preference to determine whether the decrease in SCM intake may account for the decreased SCM CPP. The analysis revealed that in WT mice, there was a significant positive correlation between SCM intake and side preference ($r = 0.71$, $p = 0.05$), however, no relationship exists in α -syn KO mice ($r = 0.47$, $p > 0.05$) indicating that the amount of SCM intake was not the critical factor influencing the lack of SCM preference. This idea is also supported by the cocaine CPP data, where all mice received the same amount of cocaine, but α -syn KO mice did not show a preference to the cocaine-paired environment. The results of the correlational analysis have been added to the results section of the paper.

3. While the effects on the Barnes Task are interesting, the authors fail to address how this is linked to a-syn mediated VTA function. As the underlying hypothesis is that a-syn in the VTA is critical for behavioral strategies used in these tasks, demonstration that Barnes Maze performance/training alters endogenous a-syn function (while a full electron microscopy assay of vesicle formation and connectivity is unnecessary, authors should include at minimum effects of Barnes Maze reversal learning on a-syn expression, and subsequent changes in ALIX and CDH6 expression) to draw these conclusions.

The main purpose of completing the Barnes Maze test was to be a control measure to determine whether the α -syn KO mice had spatial memory deficits that may contribute to the lack of cocaine conditioned place preference. The unexpected additional finding of increased perseverative errors from the α -syn KO mice were not attributed to alpha-synuclein function in the VTA, but rather effects in specific functional pathways (hippocampus vs. prefrontal cortex). The manuscript has been edited to more clearly define and explain these findings.

4. A major concern in constitutive knockout animal models is differentiating the role of alpha-synuclein in neural development versus its role in the adult brain, identifying a brain-region specific effect of a-syn, and generating a subsequent causal relationship. For example, in Fig 5B left, saline-treated a-syn KO animals show baseline differences in MVB in TH+ dendrites compared to WT controls. Moreover, as both the Barnes Maze and CPP tasks are also dependent on the hippocampus, it is possible that the identified effects on behavioral performance are being mediated independently of the VTA (or at least being mediated by hippocampal glutamatergic projections onto TH+VTA neurons, as is suggested by Fig. 4). Therefore, it is difficult to draw a causal relationship between alpha-synuclein function and alterations in VTA connectivity/molecular response to cocaine. This can be addressed in two ways: a. Does over-expression of a-syn in the VTA of a-syn KO animals restore behavioral performance and cocaine-induced effects on ALIX and CD63 ? b. Does deleting a-syn in the VTA of adult WT animals induce the same response pattern to cocaine as seen in the a-syn KO animals?

The reviewer makes an excellent point on determining the specificity of α -syn function in the VTA and these are studies we are currently conducting. Although the constitutive knockout used in the current study cannot decisively determine the extent of VTA contribution to the behavioral changes described in the manuscript, it provided valuable novel information about the necessity of α -syn for cognitive flexibility and MVB formation in dopamine neurons.

Minor points from Reviewer #2:

1. The author's claim that they have demonstrated an interaction between alpha-synuclein and ALIX is inaccurate. Their data demonstrates a potential relationship with ALIX, however, to demonstrate a direct interaction between these 2 proteins, the authors will need to provide data from a co-immunoprecipitation, FRET, or Yeast-2-Hybrid experiment. Thus, the language in the abstract and discussion should be changed to address this.

The abstract and discussion have been edited to remove any description of an interaction between α -syn and ALIX.

2. Total locomotor data for CPP experiments should be reported.

The CPP experiments report a ratio of the total time (locomotor time + resting time) in each side (cocaine/SCM paired over saline/water paired side).

3. Throughout the manuscript, the authors reference CPP results as a measure for motivation. However, the conditioned place preference task measures associative memory for rewarding (or aversive stimuli), not motivation. This should be addressed throughout the manuscript and discussion.

The "lack of motivation for hedonic stimuli in alpha-synuclein KO mice" addressed in the discussion refers to the decreased intake of SCM and not the lack of side preference exhibited in

these mice. However, the manuscript has been revised to use more precise language for both conditioned place preference and motivation.

4. *The dose of cocaine used for CPP experiments should be noted in the methods section.*

The dose of cocaine used for CPP studies was reported repeatedly in the original manuscript in both the methods section under the heading “Drug Administration” and again in the “Place Preference Conditioning” section.

5. *The results section often references data without an appropriate reference for the corresponding figure or statistics and should be re-written for clarity.*

The results section was revised to ensure inclusion of all figure and statistical references.

Suggested changes from Reviewer #3 (Remarks to the Author):

1. *The authors do an excellent job of outlining specific changes that occur in α -syn expression within the VTA following cocaine exposure. However, it seems the relationship between behavior expressed in Figure 1 (cocaine CPP) / Figure 2 (Barnes Maze) and their in depth analysis of α -syn changes in the VTA following cocaine is not entirely clear. There are attempts to explain the relationship throughout the discussion, but without direct manipulations of VTA α -syn in cocaine treated animals, or at least correlations between VTA α -syn change and changes in behavior, it is difficult to conclude that the behavior is related to these well-outlined changes in α -syn. Wouldn't α -syn KO be through the whole brain in KO mice, and α -syn expression increase through the whole brain in cocaine-treated animals? The discussion pertaining to not knowing what α -syn is doing to release of Glutamate, GABA, and dopamine within the VTA reinforces this notion.*

We thank the reviewer for their insightful comments and agree that the original manuscript was unclear and overreaching in regards to attributing the behavioral changes seen in the alpha-synuclein knockout mice to the observed changes in alpha-synuclein distributions reported in the VTA. The manuscript has been revised to indicate that the behavior changes may be a functions of the lack of alpha-synuclein in other brain regions and the link between the behavioral changes and alpha-synuclein function in the VTA have been toned down.

2. *Perhaps related to comment #1, this manuscript present a number of findings, each of which would likely stand on its own with respect to independent publications, but together form a hodgepodge of stories that is difficult to tie together (and ultimately link to behavior as in comment #1). How do pre- and post-synaptic changes tie in to the behavior.*

The research in this manuscript describes a number of novel findings and includes a thorough account of various *in vitro* and *in vivo* techniques used to determine cocaine-mediated changes in alpha-synuclein. At this point, the reported pre- and postsynaptic changes of alpha-synuclein can only explain the observed behavioral change to the extent of previously reported or known contributions of the VTA to addictive behaviors. We have edited the manuscript to make these distinction clearer and more precise. While we would very much like to have more conclusive answers to what the alpha-synuclein changes in the VTA mean in respects to addictive behaviors, we are still at the initial stages of this line of research. However, there are currently a number of ongoing studies in our laboratory to continue discerning the role of alpha-synuclein in the VTA specifically.

3. 72 hours post cocaine was chosen as a time that is presented not as a withdrawal time, but a time for cocaine to no longer be present. Care was taken to show little to no somatic withdrawal signs. There was no mention (one way or the other) or control for psychological signs of withdrawal that may precede somatic signs at this time point (e.g., anxiety-like behavior)?

We thank the reviewer for this excellent suggestion and have completed studies examining anxiety in both wildtype and alpha-synuclein knockout mice 72 hours after their last cocaine injection. The findings have been included in the methods section and supplemental figure 2. To test the potential emotional aspects of withdrawal, WT and α -syn KO mice that were naïve to cocaine (sal) or in withdrawal (rep coc -) were tested on a battery of anxiety measures including the elevated zero maze, assessed on their position (periphery vs. center) in an open field apparatus, and measured the rate of defecation in a 30 minute test period. Indeed, WT mice showed an increase in anxiety-like behaviors 72 hours after repeated cocaine administration as compared to saline controls (Suppl. Fig 2). These mice spent less time in the open areas of the zero maze ($F(1, 11) = 8.26, p = 0.017$) and made fewer attempts to enter the open areas (supplemental figure 2; $F(1, 11) = 9.06, p = 0.013$). In addition they had an increased number of fecal pellets emitted in a 30 minute test period ($F(1, 11) = 28.21, p = 0.001$). However, there were no significant differences between WT mice in withdrawal and cocaine-naïve mice in the duration of time these mice spent in the periphery (as compared to the center) of the open field apparatus ($F(1, 11) = 0.10, p = 0.76$).

It was more difficult to determine anxiolytic effects of cocaine withdrawal in the α -syn KO mice because these mice displayed higher levels of anxiety in general as seen in the open field where α -syn KO mice, regardless of cocaine exposure, show increased time in the periphery of the open field as compared to WT mice (supplemental figure 2B; $F(1,23) = 6.26, p = 0.021$). However, there is some evidence that cocaine withdrawal also increases anxiety in these mice in that α -syn KO mice in withdrawal spend less time in the open areas of the zero maze ($t(1,11) = , p = 0.0$) and have greater rates of defecation ($t(1, 11) = , p = 0.0$) as compared to cocaine naïve α -syn KO mice. There were no significant differences in ambulatory speed by prior cocaine exposure ($F(1, 23) = .0009, p = .934$), but the α -syn KO mice were generally slower than their WT counterparts ($F(1,23) = 8.45, p = 0.009$).

Minor point from Reviewer #3

1. Figure 4B labels. Asyn is used instead of α -syn (greek letter) like other figures.

The labels in Figure 4B have been changed to match the other figures.

REVIEWERS' COMMENTS:

Reviewer #2 (Remarks to the Author):

The manuscript from Trubetckaia et al. outlines the effects of alpha-synuclein (a-syn) on VTA circuitry and cocaine-associated behaviors. Specifically, this work uses a KO mouse model to assess the role of a-syn on various behavioral tasks and synaptic remodeling in the VTA in response to cocaine. First, the authors demonstrate that a-syn KO animals have impaired cocaine-induced CPP and reversal learning in the Barnes Task, but spared cocaine sensitization. Lastly, using electron microscopy, authors demonstrate cocaine-induced alterations in VTA synaptic connectivity via a-syn-mediated protein expression. Methodologically, authors have provided sufficient detail with regard to experimental methods and statistical analyses for external reproducibility.

In this resubmission from Trubetckaia et al. the authors address several, if not all, of the concerns raised (a portion of which were caused by oversights on my part). The results of these studies will be of great interest to the readership and add to the body of literature linking a-syn to neuronal function in the midbrain and establish further links between cocaine-induced adaptations and a-syn function. I have listed any additional concerns as "Additional Concerns" below. Moreover, responses to my previous concerns are addressed below as "Responses to Concerns":

Additional Concerns:

1. The term "addiction" used throughout the manuscript should be replaced with "substance use disorder".

Responses to Concerns:

Major

1.

A. All electron microscopic studies utilized four mice per experimental group (see above).

This is clearly an oversight on my part and I apologize.

B. The alpha-synuclein KO coc and rep coc – groups have been added (see above).

These experiments have been conducted and data incorporated/discussed appropriately, the authors have addressed this fully.

2. The requested information for the total SCM consumption during conditional sessions was already reported in Figure 2C (daily intake in ml) in the original manuscript. However, we greatly appreciate the reviewer's recommendation for a correlational analysis between SCM intake and side preference to determine whether the decrease in SCM intake may account for the decreased SCM CPP. The analysis revealed that in WT mice, there was a significant positive correlation between SCM intake and side preference ($r = 0.71$, $p = 0.05$), however, no relationship exists in a-syn KO mice ($r = 0.47$, $p > 0.05$) indicating that the amount of SCM intake was not the critical factor influencing the lack of SCM preference. This idea is also supported by the cocaine CPP data, where all mice received the same amount of cocaine, but a-syn KO mice did not show a preference to the cocaine-paired environment. The results of the correlational analysis have been added to the results section of the paper.

With regard to the correlation between SCM consumed and CPP score, although differences in consumption (leading to a lack of CPP) would be concerning, similar findings in cocaine-induced CPP with equal doses of cocaine lessen interpretation concerns. In addition, the lack of correlation between SCM consumed and SCM CPP in KO animals is an interesting finding. Regarding the overall consumption, I apologize for the misunderstanding regarding Fig. 2C, as I understood that to be baseline consumption in a separate experiment (not consumption during CPP training that was tested in Fig. 2D). It would be helpful to clarify this in the figure legend.

3. The main purpose of completing the Barnes Maze test was to be a control measure to determine whether the a-syn KO mice had spatial memory deficits that may contribute to the lack of cocaine

conditioned place preference. The unexpected additional finding of increased perseverative errors from the α -syn KO mice were not attributed to alpha-synuclein function in the VTA, but rather effects in specific functional pathways (hippocampus vs. prefrontal cortex). The manuscript has been edited to more clearly define and explain these findings.

The authors addressed this fully and the additional discussion greatly adds to the interpretation of the presented data.

4. The reviewer makes an excellent point on determining the specificity of α -syn function in the VTA and these are studies we are currently conducting. Although the constitutive knockout used in the current study cannot decisively determine the extent of VTA contribution to the behavioral changes described in the manuscript, it provided valuable novel information about the necessity of α -syn for cognitive flexibility and MVB formation in dopamine neurons.

The authors do not address this experimentally, however, changes in the discussion addressed elsewhere are sufficient.

Minor:

1. The abstract and discussion have been edited to remove any description of an interaction between α -syn and ALIX.

The authors have addressed this fully.

2. The CPP experiments report a ratio of the total time (locomotor time + resting time) in each side (cocaine/SCM paired over saline/water paired side).

The total locomotor activity has been reported during training and testing for cocaine-induced CPP (Fig. 1A), however the comparable data in SCM CPP is missing and should be included (total ambulatory time) in Fig. 2.

3. The "lack of motivation for hedonic stimuli in alpha-synuclein KO mice" addressed in the discussion refers to the decreased intake of SCM and not the lack of side preference exhibited in these mice. However, the manuscript has been revised to use more precise language for both conditioned place preference and motivation.

The authors have addressed this fully.

4. The dose of cocaine used for CPP studies was reported repeatedly in the original manuscript in both the methods section under the heading "Drug Administration" and again in the "Place Preference Conditioning" section.

This is clearly an oversight on my part, and I apologize. The authors had already addressed this fully.

5. The results section was revised to ensure inclusion of all figure and statistical references.

The authors have addressed this fully.

Reviewer #3 (Remarks to the Author):

The language pertaining to results in the VTA and the relationship to behavioral measures have been generally well revised and the toning down of the relationship is appreciated. It was surprising that the behavioral results pertaining to anxiety-like behavior received little consideration in the discussion given that some of the effects were as robust, if not more, than other behavioral measures. Emotional withdrawal signs are mechanistically linked to addiction vulnerability or relapse phenotypes, so this seems very interesting to be ignored.

A remaining suggestion would be to move methods for the measures of anxiety-like behavior to the "behavioral measures" section of the methods rather than before it, and to put the results of those behaviors in the results section (rather than the methods). The current organization made it difficult to find and having result within the methods section was hard to follow.

Reviewer #2 additional requests

Additional concern point 1: The term addiction has been replaced with substance abuse disorder throughout the manuscript.

Major Concern point 2: The text in the figure legend was changed to indicate that the daily intake of water and SCM was recorded during the conditioning trials to avoid any confusion.

Minor Concern point 2: The total locomotor activity has been added to Figure 2.

Reviewer #3 additional requests

The methods for the measures of anxiety-like behavior has been moved to the “Behavioral Measures” section of the methods as requested. However, I was unable to move the results of those behaviors to the results section because of word limitations. Instead they remain with the descriptive information already present for the somatic signs of withdrawal in the methods section and as a supplemental figure.